# Modelling Municipal Cost Inefficiencies in the Frances Baard District of South Africa and Their Impact on Service Delivery

Brian Tavonga Mazorodze 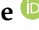

Faculty of Economic and Management Sciences, Department of Accounting and Economics, Sol Plaatje University, Kimberley 8301, South Africa; brian.mazorodze@spu.ac.za

**Abstract:** Section 195(b) of South Africa's Constitution calls for the efficient use of resources in public administration, while the White Paper on Local Government similarly emphasizes the efficient provision of basic services such as refuse collection and water. Despite these policy commitments, South African municipalities continue to be characterized by financial mismanagement and poor service delivery. In light of the limited empirical evidence on this issue, this study pursues two objectives. First, it estimates the levels of cost inefficiency in four local municipalities within the Frances Baard District from 2006 to 2023. Second, it determines how cost inefficiencies affect service delivery, focusing on water and refuse collection. Using a stochastic frontier analysis, several key results are confirmed. First, on average, the four municipalities are found to have spent 17.23% above the minimum cost required to deliver existing services. Second, service delivery is found to have been 23% lower than its potential. Third, operating costs and contracted services are found to have been key drivers of wasteful expenditure. Lastly, cost inefficiencies are found to have culminated in poor service delivery. Against this background, municipalities are urged to consider cutting non-essential operational spending such as entertainment and travel allowances, coupled with improved accountability on contracted services.

**Keywords:** municipal cost inefficiency; local municipalities; service delivery



## 1. Introduction

South Africa's local municipalities have been in the spotlight in recent decades for financial mismanagement, maladministration and poor service delivery. According to the audit outcomes contained in the 2021 state of local governance finances and financial management report, most local municipalities were in a dire financial position. The same report indicates that 25 municipalities could not provide sufficient documentation for the financial information disclosed in their financial statements, while irregular expenditure stood at R21.9 billion in 2020/21. This maladministration has, in most instances, culminated in poor service delivery. In 2021, for instance, and for the first time in a democratic South Africa, the High Court had to intervene in one of the financially distressed municipalities following what the Auditor General's 2020/21 report described as a service delivery crisis in the municipality. As of 30 June 2021, 38 municipalities were under some intervention either through mandatory or discretionary, in terms of Section 139 of the Constitution, which empowers provincial executives to intervene in municipalities that fail to fulfill their constitutional or legislative obligations, including approving budgets or providing basic services.

Despite the policy relevance of this maladministration, very few empirical studies have paid attention to the cost management of local municipalities and its effect on service delivery. The majority of the existing studies have qualitatively explored challenges

facing local municipalities without necessarily quantifying cost efficiency and its effect on service delivery. These studies include Kanyane (2014), Laubscher (2012), and more recently Mamokhere and Thusi (2024), Mashabela and Thusi (2024), Mamokhere (2024) and Ralinala et al. (2024). The few available studies that have quantified the efficiency of local municipalities in utilizing public resources include Monkam (2014) and Adedeji Amusa and Fadiran (2024). While these studies provide valuable insights into the efficient use of public resources by municipalities, they do not explicitly focus on cost efficiency, defined as the ability of local municipalities to deliver basic services at the minimum cost.

The current study therefore differs from these studies by measuring the cost inefficiency of local municipalities and estimating its effect on service delivery. The specific aim of the study is to measure the level of cost inefficiency across municipalities and establish its impact on service delivery with a specific focus on water and refuse collection. The primary objective is to understand the extent to which cost inefficiency of municipalities culminates in poor service delivery. The research hypothesis raised in this study is that municipalities characterised by wasteful expenditure are associated with poor service delivery. The empirical evidence is drawn from four local municipalities belonging to the Frances Baard District of the Northern Cape province of South Africa, where concerns of financial mismanagement and poor service delivery have been raised in recent years. Filling this empirical gap is important considering that cost inefficiency in local municipalities has direct implications for public sector accountability, optimum allocation of public resources, and service delivery outcomes. In a context where municipalities face increasing fiscal pressure and public dissatisfaction due to poor service provision, quantifying cost inefficiency allows for evidence-based interventions that target the root causes of wasteful expenditure. In addition, unlike qualitative and explorative studies, an empirical approach provides measurable benchmarks that can inform policy decisions and performance monitoring from a perspective of local governance. By explicitly linking cost inefficiency to service delivery outcomes, this study has the potential to guide reforms aimed at achieving the constitutional and policy mandates of efficient and equitable local governance.

The study contributes to the body of knowledge by exclusively focusing on local municipalities situated in a desert economy. The study area is particularly located in a dry region where economic activity is heavily concentrated in diamond mining. This provides an interesting case as local governments in such areas face a dual challenge of managing the volatility of commodity-dependent revenue streams and navigating the governance complexities that accompany rent-seeking behaviors endemic to resource-rich contexts. Frances Baard particularly exemplifies a broader phenomenon observed in resource-dependent economies where local governments often struggle to balance fiscal stability with the provision of essential services. Price shocks in commodity markets often erode municipal revenues, exacerbating inefficiencies in service delivery. Understanding how these dynamics play out in the Northern Cape, where weather conditions further complicate economic certainty, potentially sheds light on broader governance challenges faced by municipalities in similar economic settings. The literature additionally highlights the governance challenges affecting municipalities in mining regions where local elites may capture public resources and distort expenditure patterns, which undermines cost efficiency. While the findings are geographically specific, they carry broader implications for Sub-Saharan Africa, where several municipalities grapple with the governance challenges associated with resource dependency and volatility of fiscal revenues.

Using stochastic frontier analysis, the study finds that, on average, municipalities spent 17.23% more than the minimum cost required to deliver existing services, while service delivery performance—specifically in water provision and refuse collection—was 23% below its potential. The analysis identifies high operating costs and spending on

contracted services as the main drivers of cost inefficiency, which in turn contributed to poor service delivery outcomes. In light of these findings, the study recommends that municipalities reduce non-essential expenditures such as entertainment and travel allowances and enhance accountability in the management of contracted services.

The remainder of the study is structured as follows. Section 2 provides theoretical and empirical literature. Section 3 outlines the materials and methods used, including the data sources, variables, and the stochastic frontier analysis technique employed to estimate cost and service delivery inefficiency. Section 4 presents and interprets the empirical results, highlighting key findings on expenditure patterns and service delivery outcomes. Section 5 provides a discussion of the results, linking the empirical evidence to the existing literature and South Africa's local governance landscape. Section 6 provides limitations of the study, while Section 7 concludes the study, offers policy recommendations, and suggests directions for future research.

## 2. Literature Review

This section provides theoretical lenses through which municipal inefficiencies can be examined, and it situates the discussion within broader debates in public finance, governance, and institutional economics.

### 2.1. Theoretical Literature

The notion that public sector actors may not always act in the best interests of citizens finds fertile ground in the public choice theory, which is largely credited to Buchanan (1983). In the main, this theory illustrates how politicians and bureaucrats, much like private sector agents, are driven by self-interest rather than public welfare. In the municipal context, this manifests through practices such as political patronage, vote-seeking behavior, and rent-seeking activities. Politicians may redirect resources to politically connected contractors, inflating procurement costs without commensurate improvements in service delivery. In Frances Baard, reports of inflated project costs and politically connected tender allocations (Molatlhwa, 2021) crudely demonstrate how electoral motives distort financial priorities, compromising cost efficiency.

A parallel narrative unfolds through the lens of principal-agent theory (Rees, 1985). The framework articulates the inherent conflicts between elected officials (principals) and municipal managers (agents). Municipal managers, who are entrusted with implementing policies and managing budgets, may engage in opportunistic behavior that erodes cost efficiency. This problem is particularly prevalent in municipalities where procurement irregularities and excessive administrative expenditures reflect a breakdown in oversight mechanisms. Emam and Govender (2024) contend that without stringent monitoring and effective incentive structures, municipal managers can manipulate budgets to their advantage, perpetuating inefficiency under the guise of operational exigencies.

If one considers the public sector as an environment where monopolistic tendencies are tolerated, X-Inefficiency theory, developed by Leibenstein (1966), becomes particularly relevant. Unlike profit-driven firms that face competitive pressures to minimize costs, municipal departments operate in insulated environments where inefficiency can persist with limited checks and balances. Leibenstein (1966) particularly suggests that without the threat of competition, public sector entities may exhibit managerial slack, operational waste, and bureaucratic inertia. In the context of local municipalities, procurement inefficiencies, coupled with excessive payroll costs, may resemble this X-inefficiency. In line with this proposition, several scholars, including Matlala et al. (2023), have argued that efforts to rationalize operations and reduce costs tend to encounter managerial resis-

tance (which perpetuates inefficiencies) in municipalities where public sector unions wield significant power.

Unlike Leibenstein (1966), Oates (1972) advances the fiscal federalism theory. In this theory, decentralization can either enhance or impede resource allocation efficiency, depending on the capacity of local governments to manage their fiscal responsibilities. In South Africa, fiscal decentralization has been a double-edged sword. While municipalities such as those in Frances Baard are granted autonomy over budgetary decisions, weak financial oversight and limited administrative capacity create fertile ground for inefficiency. Mishi et al. (2022) document how poorly executed infrastructure projects, funded through intergovernmental grants, often exceed budget estimates without delivering the intended service outcomes. In such cases, the decentralization of fiscal authority exacerbates rather than mitigates cost inefficiencies.

Last but not least is the resource dependence theory, conceptualized by Pfeffer (1987). According to this theory, municipalities in natural resource regions that rely heavily on external funding, whether from national government grants or donor contributions, may find themselves constrained either by the conditions attached to these funds or by exogenous events such as commodity price shocks. With respect to the former, in cases where the equity share constitutes a significant proportion of the municipal budget, conditional grants can inadvertently distort spending priorities. Van der Waldt (2015) particularly observes that when funds are earmarked for specific projects, municipal managers may prioritize compliance with external funding guidelines over cost-effective service delivery. Similarly, negative commodity price shocks may constrain the national budget and consequently distort the revenue and expenditure patterns of local municipalities. Therefore, the dependence on external resources can inadvertently entrench inefficiencies by exposing municipalities to external shocks (in the case of commodity price movements) and creating a culture of compliance rather than efficiency (in the case of externally funded projects).

Despite providing different mechanisms, these theories have a common denominator: cost inefficiency in municipal service delivery is rooted in institutional, political, and structural factors. In this study, we reduce the sources of cost inefficiencies to cost management, which, in turn, based on X-inefficiency, principal-agent theory, and the public choice theory, may reflect either lack of incentives by public officials to rationalize their operations or systematic corruption that siphons public resources through irregular operational expenditure and poor oversight on contracted services.

*2.2. Empirical Literature*

The literature on municipal cost inefficiencies and their implications for service delivery is quite extensive. In the main, it comprises studies that have primarily measured cost efficiency of municipalities using either the Stochastic Frontier Approach (SFA) or the Data Envelopment Approach (DEA). Although the results vary across countries, a common finding is that municipalities operate inefficiently, and inefficiency often culminates in poor service delivery. Adedeji Amusa and Fadiran (2024) provide a critical examination of service delivery efficiency across 213 local and metropolitan municipalities in South Africa, employing a partial frontier efficiency analysis (PFEA). Focusing on electricity, water, sewerage, and waste removal services, their study reveals variability in municipal efficiency, with electricity services demonstrating relatively higher efficiency rates compared to waste removal. Their findings additionally challenge conventional wisdom, indicating that smaller urban municipalities exhibit higher efficiency levels than larger metropolitan areas. Although their results provide a compelling case for targeted policy interventions aimed at mitigating efficiency gaps, their analysis focused on technical inefficiency. In a context

where reports by the Auditor General have recurrently pointed to irregular expenditures across municipalities, it might be more appropriate to rather focus on cost inefficiency.

Salsabila et al. (2025) contribute to the empirical discourse by adopting a Value for Money (VFM) framework to assess financial performance in Palopo City, Indonesia. Their study demonstrates the importance of aligning budgetary allocations with principles of economy, efficiency, and effectiveness. In particular, their analysis documents the challenges of translating budgetary inputs into tangible service delivery outcomes, a theme resonant with the findings of Adedeji Amusa and Fadiran (2024). While this conclusion is important from a policy perspective, it addresses efficiency by implication. It might be necessary to directly confront efficiency levels of municipalities given the availability of inputs, outputs and expenditure data. Expanding on the efficiency discourse directly, Rella et al. (2025) employ a two-stage DEA to assess waste management efficiency in 147 Italian municipalities. Their findings reveal a medium-high level of waste management efficiency. They additionally find the negative impact of unemployment on municipal performance, having broader implications for service delivery outcomes. This is particularly relevant to the South African context, where poor service delivery is often linked to municipal inefficiencies.

In the Dutch context, Blank and van Heezik (2025) apply a locally least squares frontier method to examine cost efficiency in youth care purchasing policies across 352 municipalities. Their results indicate significant heterogeneity in cost efficiency, with open house outsourcing and framework contracts emerging as critical determinants of efficient procurement practices. Their study aligns with the broader literature on public expenditure efficiency, which emphasizes the role of institutional arrangements and contracting practices in shaping municipal performance. The contracting practices are particularly relevant to the South African context, given how rent-seeking and corrupt practices are often associated with contracted services. In South Korea, Min and Lee (2025) explore the implications of contract-based employment for service delivery efficiency. Contrary to allegations often levelled against municipal management in South Africa, their study demonstrates that contract workers, particularly in doorstep health services, contribute to improved service delivery outcomes. Their findings, however, additionally suggest potential trade-offs between cost savings and service quality, an aspect that warrants further investigation in the context of South African municipalities grappling with resource constraints and service delivery backlogs.

Kim and Kang (2025) offer a critical perspective on administrative intensity and its fiscal implications for South Korean municipalities. Their study presents a curvilinear relationship between bureaucratic structures and fiscal performance, challenging conventional narratives that frame bureaucracy solely as a fiscal burden. Instead, they argue for an optimal level of administrative intensity that balances efficiency and capacity, a notion particularly pertinent to municipalities in the Frances Baard District, where administrative and operational inefficiencies persist. Gbambegu Umar et al. (2025) extend the literature on financial management by examining the mediating role of political interest in Ghanaian municipalities. Using a structural equation modeling approach, the study reveals that political interest significantly mediates the relationship between internal control systems and financial management. This finding is instructive for understanding the broader political economy of service delivery in South Africa, where political patronage and interest alignment can substantially affect municipal resource allocation and governance outcomes.

Óskarsson et al. (2025) focus on waste management efficiency in Iceland, emphasizing the interplay between socioeconomic factors and service delivery outcomes. Their DEA analysis demonstrates the importance of contextual variables such as population size, income levels, and rural-urban ratios, which significantly shape municipal efficiency. These

insights hold particular relevance for South African municipalities operating in resource-constrained and geographically diverse setups. Phahlamohlaka and Mpungose (2025) focus on supply chain management (SCM) practices and their impact on service delivery in local municipalities. Their qualitative analysis highlights the potential of collaborative planning and demand forecasting to enhance operational efficiency. However, persistent challenges such as limited technological integration and inadequate staff training are found to impede the full realization of SCM benefits. These findings echo those of Adedeji Amusa and Fadiran (2024), who emphasize the need for capacity-building initiatives to bridge efficiency gaps in South African municipalities.

Barbosa et al. (2025) examine the interactions between efficiency, productivity, and recycling policies in Brazilian municipalities. Using a dynamic slacks-based measure (DSBM) model, the study identifies significant inefficiencies in municipal solid waste services, particularly in the context of recyclable materials recovery. This finding has implications for South African municipalities, where recycling initiatives are increasingly prioritized but remain constrained by limited infrastructure and policy support. In Mugabe et al. (2025), a critical analysis of political patronage and its impact on service delivery is provided in Kabale Municipality, Uganda. Their study demonstrates the detrimental effects of political favoritism on resource allocation and service equity, a theme that resonates with the South African context, where political dynamics often shape municipal service delivery priorities. It is clear from this literature that studies measuring cost efficiency of local municipalities in South Africa and examining its impact on service delivery are scanty at best.

## 3. Materials and Methods

This section provides the methodology of the study. It starts by describing and justifying the study area as well as the measurement of key variables before specifying the empirical models and justifying the estimation strategy.

### 3.1. Study Area

South Africa's system of local government comprises 257 municipalities that fall into three categories in terms of the Constitution: Category A (metropolitan municipalities), which have exclusive municipal executive and legislative authority in their areas; Category B (local municipalities), which share authority with district municipalities; and Category C (district municipalities), which coordinate development and service delivery across multiple local municipalities. Local municipalities, particularly those in smaller towns and rural areas, are often faced with persistent challenges related to financial management and efficient service provision. This study focuses on Category B municipalities within the Frances Baard District, aiming to assess the extent of cost inefficiency and its implications for service delivery outcomes with a specific focus on water provision and refuse collection. The Frances Baard District is purposefully selected due to its economic and administrative significance, as it is home to Kimberley, the provincial capital of the Northern Cape. Figures 1 and 2 show the provincial and district demarcations of South Africa, respectively. In Figure 1, the Northern Cape is represented by the dark green colour. In Figure 2, Frances Baard, situated in the Northern Cape province, is similarly represented by the dark green colour.

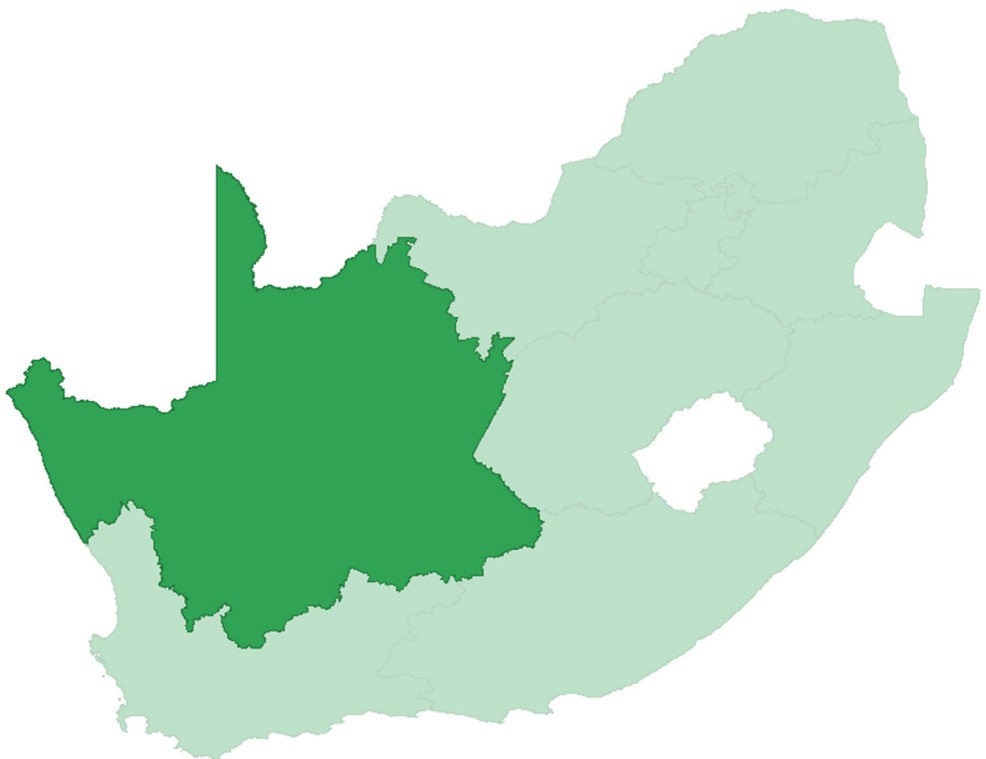

**Figure 1.** The Northern Cape Province.

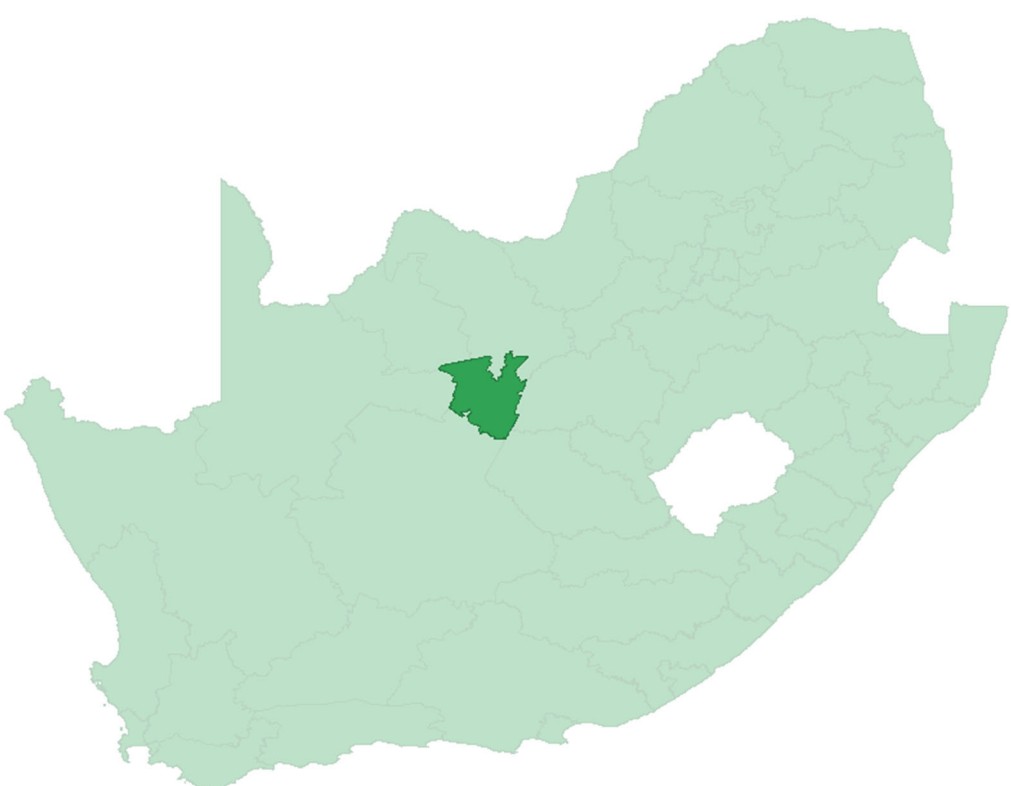

**Figure 2.** Frances Baard District. Source: Own computation.

The selection of municipalities in the Northern Cape Province is a deliberate choice based on structural economic characteristics and governance dynamics that typify resource-dependent, desert economies. The Northern Cape is a dry province known for its diamond mines. In regions where diamond mining constitutes the primary economic activity, municipal revenue generation is inextricably linked to exogenous commodity price shocks.

This dependence introduces fiscal vulnerabilities as fluctuations in commodity prices can rapidly destabilize local revenue streams, undermining the financial base necessary for cost-effective service delivery. In particular, Frances Baard, being a major diamond-mining region, is characterized by cyclical booms and busts driven by global market demand and geopolitical factors. This cyclical nature creates unique governance challenges as municipalities ought to navigate periods of resource windfalls and revenue contractions. Such volatility, coupled with limited economic diversification, makes this region unique and interesting for modelling municipal cost inefficiencies and establishing their impact on service delivery.

In addition, mining regions often operate within governance frameworks that are particularly susceptible to rent-seeking behavior, where those in power may have disproportionate influence over resource allocation, distorting expenditure patterns and exacerbating cost inefficiencies. In other words, the study area's fiscal dependence on a volatile mining sector and institutional weaknesses that often accompany resource-dependent municipalities provide conditions conducive to municipal inefficiency and poor service delivery. While geographically specific, the findings have broader relevance for municipalities across Sub-Saharan Africa where similar economic and governance dynamics prevail. Consequently, the study advances the discourse on municipal governance in resource-dependent contexts, providing empirically grounded recommendations for mitigating cost inefficiencies in settings where economic fortunes are closely tied to commodity cycles.

Frances Baard district comprises municipalities whose governance and service delivery challenges have been in the spotlight in recent years. In 2021, for instance, according to a report by the Parliamentary Monitoring Group (PMG), three of the four local municipalities in this district—namely Dikgatlong, Magareng, and Phokwane—were in distress while the fourth, Sol Plaatje, was unable to pay its Eskom debt and faced declining service delivery. These governance, financial and service delivery challenges make Frances Baard District ideal for analyzing municipal cost inefficiency and its drivers.

### 3.2. Measuring Cost Efficiency of Municipalities

While there are different types of efficiency, this study primarily focuses on cost efficiency as a key factor that affects service delivery. Focusing on cost efficiency is particularly relevant in South African municipalities given the increasing pressure on municipalities to do more with less amidst declining fiscal transfers and escalating operational costs. Cost efficiency analysis allows the identification of potential cost-saving measures without compromising service quality, thus serving as a critical mechanism for enhancing financial sustainability and facilitating accountability.

Empirically, cost inefficiencies and their sources can be estimated broadly using either the Stochastic Frontier Analysis (SFA) of Meeusen and van den Broeck (1977) and Aigner et al. (1977) or the Data Envelopment Analysis (DEA) of Charnes et al. (1978). The DEA is, in the main, deterministic and uses programming methods to construct a frontier from which deviations of actual costs are referred to as cost inefficiencies. The SFA, on the contrary, separates random noise from cost inefficiencies, making it superior to the DEA as its stochastic nature makes it less susceptible to exaggerating cost inefficiencies. In addition, municipal data are usually characterized by measurement error in reporting, something that the SFA is designed to handle. It is for these reasons that the stochastic frontier analysis was preferred over the data envelopment approach for this analysis. In the main, the stochastic cost frontier model estimates the minimum possible cost beyond which municipalities are deemed to be cost inefficient. Intuitively, in the context of local governance, the estimated minimum cost level would represent necessary spending while actual costs above this estimated minimum cost level would signify wasteful expenditure.

### 3.3. Measuring Service Delivery

While municipalities, in the main, deliver a wide range of services, this study focuses on water provision and refuse collection. Electricity and housing are deliberately excluded. The exclusion of electricity as a service delivery measure in this study is justified on several grounds. Firstly, the responsibility for electricity distribution in South Africa is not uniformly handled by municipalities. While some municipalities are directly involved in electricity distribution, others rely on Eskom, the national electricity supplier. This fragmented structure creates inconsistencies in the provision of electricity as a municipal service, making it less suitable for a comparative analysis across local municipalities within the Frances Baard District. Secondly, data availability and consistency present significant challenges. Data on electricity provision are often aggregated at the national or provincial level, particularly in areas where Eskom is the sole distributor. In contrast, data on refuse collection and water provision are more consistently available and directly attributable to municipal operations, allowing for more accurate measurement and comparison of cost inefficiency. In addition, municipalities typically have greater control over refuse collection and water services as these functions fall entirely within their constitutional mandates under the Municipal Systems Act and the Water Services Act. Electricity distribution, however, is subject to external regulation and operational decisions by Eskom, limiting the ability of municipalities to influence service quality and efficiency. Additionally, the cost structures of electricity provision differ substantially from those of water and refuse collection, given the capital-intensive nature of electricity infrastructure and the regulation of tariffs at the national level. Including electricity in the analysis would therefore introduce variability in cost structures that could confound the interpretation of cost inefficiency estimates.

The exclusion of housing as a service delivery measure, on the other hand, is premised on its distinct administrative structure and the limited role of local municipalities in its provision. In South Africa, the responsibility for housing delivery primarily resides with provincial and national governments, as stipulated under the Housing Act (Act No. 107 of 1997). Municipalities are typically involved in the identification of suitable land, basic service provision (such as water and sanitation), and facilitation of housing projects, but they are not the primary implementers of housing delivery. As such, including housing as a service delivery measure would not accurately reflect the cost efficiency of municipal operations in the Frances Baard District.

In addition, the funding structures for housing projects are distinct from those of refuse collection and water provision. Housing projects are largely financed through conditional grants and subsidies allocated by provincial and national authorities, which are earmarked for specific purposes and are not subject to municipal discretion in the same way as operational budgets for refuse collection and water services. Consequently, incorporating housing in the analysis would introduce variability in funding and expenditure patterns that could obscure the assessment of municipal cost inefficiencies. Moreover, housing delivery involves long-term capital-intensive projects with extended implementation timelines, whereas refuse collection and water services are ongoing operational functions with more regular expenditure patterns. Including housing in the cost inefficiency analysis would therefore necessitate a fundamentally different methodological approach, potentially involving capital expenditure assessments and project-based cost analysis, which falls outside the scope of this study.

The decision to focus on water and refuse collection services is substantiated by documented challenges in the Frances Baard District Municipality, particularly within Sol Plaatje Municipality. These services are fundamental to public health and urban sustainability, yet they face systemic issues that impede effective delivery. Kimberley, the capital of the Northern Cape, relies on the Riverton Water Treatment Works and associated infrastruc-

ture for its water supply. However, the aging infrastructure has led to frequent water interruptions, with residents reporting prolonged periods without water. In early 2024, a compilation of 242 complaints highlighted widespread dissatisfaction, with residents citing daily hardships and violations of their constitutional right to clean water. The provincial Department of Water and Sanitation has identified Sol Plaatje Municipality as a focal point for infrastructure improvement. A Bulk Infrastructure Grant Facility has been allocated to address bulk water infrastructure challenges, focusing on projects to be implemented over seven to eight years. This initiative underscores the critical need for substantial investment in water infrastructure to mitigate service delivery issues. At the same time, refuse collection and waste management issues are persistent concerns in the region, with residents reporting irregular and inadequate waste removal services. In response, the Frances Baard District Municipality allocated a refuse collection truck valued at ZAR 1.3 million to Sol Plaatje Municipality to enhance service delivery and address backlogs. Illustratively, the district has faced criticism regarding landfill management practices. In November 2024, for instance, the region was criticised for illegal dumping at landfill sites, demonstrating systemic issues in waste management and the need for strategies to ensure compliance with environmental regulations. On account of these considerations, focusing on water and refuse collection services in Frances Baard is imperative to the extent that it allows for a targeted examination of critical service delivery areas that directly impact residents' quality of life. The documented challenges in Frances Baard particularly provide a compelling case for prioritizing these sectors when modelling service delivery as an outcome variable.

### 3.4. Data Description

The study particularly uses a micro-panel dataset comprising four category B municipalities in the Frances Baard District—Dikgatlong, Magareng, Phokwane, and Sol Plaatje—observed annually from 2006 to 2023. The choice of panel over time-series and cross-sectional data is justified by the need to account for unobserved heterogeneity (Baltagi et al., 2009). The sampling period is dictated by data availability on municipal revenue and expenditure. We draw our data from Statistics South Africa, through Quantec. From this source, revenue and expenditure data are only available from 2006 to 2023.

### 3.5. Model Specification

The model specification has two parts. The first part measures cost inefficiency of local municipalities and estimates its relevant sources. The second part uses the measured cost inefficiencies to establish their impact on service delivery. Following Arcelus et al. (2015) with modifications in the estimation procedure and the handling of endogeneity, the panel stochastic cost frontier model applied in this study for the first objective takes the following form.

$$C_{it} = \exp\{x_{it}\beta + e_{it}\} \tag{1}$$

where subscripts *i* and *t* denote municipality and year *C* is the full cost of running a municipality, *x* is a vector of two output indicators namely refuse collection and water supply proxying the scale of service delivery in line with the stochastic cost frontier literature (Kumbhakar et al., 2015), $\beta$ is a vector of cost frontier parameters to be estimated, *e* is an error term comprising the stochastic component ($v_{it}$) and the cost inefficiency term ($u_{it}$) capturing the extent to which actual costs exceed the minimum cost as indicated in Equation (2).

$$e_{it} = v_{it} + u_{it} \tag{2}$$

The inefficiency term is then specified as a dependent variable to appreciate the sources of cost inefficiencies as follows:

$$u_{it} = z_{it}\delta + w_{it} \tag{3}$$

where $z$ is a vector capturing the sources of cost inefficiency, $\delta$ is a corresponding vector of slope coefficients to be estimated, and $w$ is the error term. Vector $z$ includes the share of bulk purchases on total expenditure, the share of expenditure on contracted services, operating costs, debt impairment arising from households not paying municipal rates, and employee remuneration as a share of total expenditure. This approach is consistent with the stochastic frontier cost function methodology, where total cost is modeled as a function of outputs, while inefficiency is explained by cost management variables. Operational costs are included as they represent the recurrent expenditures that underpin service delivery. Prior studies such as Farsi and Filippini (2004) and Worthington and Dollery (2001) indicate that higher operational costs are associated with inflated expenditure structures, suggesting that municipalities with higher operating costs relative to outputs are likely to exhibit higher inefficiency. This inclusion additionally aligns with the X-inefficiency theory reviewed earlier, which posits that public sector entities without competitive pressures may exhibit managerial slack and operational waste. Hence, municipalities with higher operational costs relative to outputs are expected to be more inefficient, leading to a positive expected sign. The expected sign is therefore positive.

Debt impairment, reflecting poor debt collection and financial mismanagement, is considered a crucial source of inefficiency. As highlighted by Działo et al. (2019), higher levels of debt impairment are indicative of fiscal stress, which can divert resources away from service provision and inflate overall costs. This is also consistent with principal-agent theory, where municipal managers may engage in budgetary manipulation to obscure fiscal stress. Consequently, the coefficient on debt impairment is anticipated to be positive. Employee remuneration as a share of total expenditure is included to capture the potential effects of bureaucratic expansion on cost inefficiency. Monkam (2014) documents that elevated payroll expenses are symptomatic of inefficient administrative structures, suggesting that municipalities with higher employee remuneration relative to outputs are more likely to incur excessive costs. Thus, the expected sign is positive.

The share of expenditure on contracted services is added to assess the implications of outsourcing on cost efficiency. Balaguer-Coll et al. (2007) underscore the potential for contracted services to facilitate rent-seeking behavior, thereby increasing inefficiency. The public choice and the principal-agent theories suggest that municipalities with high shares of expenditure on contracted services and employee remuneration may experience cost inefficiencies driven by opportunistic behavior and bureaucratic expansion. Consequently, a positive coefficient is expected for contracted services, indicating that municipalities with higher contracted service expenses are more likely to be inefficient. Lastly, the share of bulk purchases on total expenditure is included to examine potential cost savings arising from economies of scale. In theory, bulk purchasing could mitigate cost inefficiencies by lowering per-unit costs through economies of scale. This expectation is consistent with resource dependence theory, where municipalities that largely depend on external funding may strategically manage resource allocation to mitigate cost inefficiencies. Therefore, a negative coefficient is anticipated.

The panel stochastic production frontier for the second objective takes the following form.

$$Y_{it} = \exp\{h_{it}\beta + \epsilon_{it}\} \tag{4}$$

where $Y$ is an output vector capturing water provision and refuse collection, $h$ is a vector capturing municipal revenue and municipal workers, while $\epsilon_{it}$ is a composite error term

which comprises the random term ($v_{it}$) and a component ($\tau_{it}$) which captures the inability of municipalities to provide a maximum level of service delivery as follows:

$$\epsilon_{it} = v_{it} - \tau_{it} \tag{5}$$

Since we hypothesize that cost inefficiencies may prevent municipalities from providing service delivery at full potential, we specify the ($\tau_{it}$) as a function of cost inefficiencies measured earlier as follows:

$$\tau_{it} = u_{it}\vartheta + \theta m_{it} + \mu_{it} \tag{6}$$

where $\vartheta$ is a slope coefficient capturing how cost inefficiencies affect the ability of municipalities to provide service delivery at full potential and $m_{it}$ is a vector of control variables namely the log of population, unemployment, the log of Gross Value Added (GVA) and functional literacy measured by the percentage of people with at least secondary school education. Guided by economic intuition, the log of population is included as larger municipalities may have greater service delivery challenges due to higher demand for services and potential congestion effects. Unemployment is considered to capture socio-economic pressures that may exacerbate service delivery constraints, as municipalities in areas with high unemployment may face higher demands for social services and limited revenue bases. Gross Value Added (GVA) serves as a proxy for the economic capacity of municipalities to generate revenue and allocate resources efficiently. Finally, functional literacy, defined as the percentage of people with at least secondary school education, is included to account for the human capital component, as higher literacy rates may correspond with better management practices and more effective governance structures in municipalities. Noteworthy is that Equation (5) assumes that municipalities' objective function seeks to maximise water provision and refuse collection from given revenue and a fixed number of municipal workers. Their ability to achieve this objective is assumed to depend on how well they manage their costs (i.e., cost efficiency) based on Equation (6).

Tables 1 and 2 present the variable description for objectives 1 and 2, respectively. All data are from Quantec.

**Table 1.** Variable Description and Classification—Objective One.

| Variable | Description | Classification |
|---|---|---|
| Total cost | Aggregate municipal expenditure | Dependent variable in the cost frontier specification |
| Water provision | Number of households with tapped water inside dwelling | Cost frontier regressor |
| Refuse collection | Number of households whose refuse is collected at least once a week | Cost frontier regressor |
| Operating costs | Operating costs as a percentage of total expenditure. This includes advertising, marketing, communication, entertainment, hire charges, insurance underwriting, printing, transport, travel and subsidies and professional memberships. | Source of cost inefficiency |
| Bulk purchases | Bulk purchases as a percentage of total expenditure | Source of cost inefficiency |
| Contracted services | Cost of contracted services as a percentage of total expenditure | Source of cost inefficiency |
| Employee remuneration | Employee remuneration as a percentage of total expenditure water | Source of cost inefficiency |

Table 2. Variable Description and Classification—Objective Two.

| Variable | Description | Classification |
|---|---|---|
| Water provision | Number of households with tapped water inside dwelling | Dependent variable in the service delivery specification |
| Refuse collection | Number of households whose refuse is collected at least once a week | Dependent variable in the service delivery specification |
| Total revenue | Aggregate municipal expenditure | Service delivery regressor |
| Workers | Total municipal workers | Service delivery regressor |
| Cost inefficiency | The extent to which actual costs are above the estimated minimum cost level | Regressor in the service delivery specification |
| Population growth | The log of total population | Control variable |
| Level of education | Functional literacy (% of total population) | Control variable |
| Unemployment | Percentage | Control variable |
| Local economic output | Log gross value added | Control variable |

*3.6. Estimation Approach*

The appropriate estimation approach was determined based on three methodological caveats which, if ignored, could lead to bias in both cost inefficiencies and slope parameters. The first caveat is unobserved heterogeneity, which, if ignored, could lead to cost inefficiencies that are contaminated with time-invariant factors specific to each municipality. The model will additionally suffer from heterogeneity endogeneity if municipal-specific factors such as management style, work ethics, and geography, among others, are correlated with frontier and inefficiency variables. The second caveat relates to the treatment of idiosyncratic endogeneity, which occurs when time-varying factors nested in $v_{it}$ and $w_{it}$ are correlated with frontier variables and drivers of cost inefficiency. Ignoring this caveat invites a bias that does not disappear even in large samples. The third aspect is the question of whether the stochastic cost frontier model should be estimated using the two-step or the one-step procedure. Traditionally, as indicated by Wang and Schmidt (2002), studies have used the two-step procedure where the cost inefficiencies are estimated in the first step and then separately linked to their determinants in the second, mostly using Tobit regressions. Wang and Schmidt (2002) have demonstrated, however, that this procedure leads to severe bias. They recommend a one-step procedure in which the stochastic cost frontier model and the inefficiency specification are simultaneously estimated using the maximum likelihood method. This is the approach used in this study.

Within the literature, stochastic frontier models have evolved significantly. Earlier models and estimation procedures include Cornwell et al. (1990) and Lee and Schmidt (1993), Pitt and Lee (1981), and Battese and Coelli (1988, 1995). In this paper, our estimation procedure follows Karakaplan (2022) and Greene (2005). These modelling approaches are preferred over their alternatives due to their ability to handle idiosyncratic endogeneity and unobserved heterogeneity, respectively. It is important to point out, however, that in their original forms, none of the modelling approaches account for both unobserved heterogeneity and idiosyncratic endogeneity. Greene's (2005) approach addresses unobserved heterogeneity but does not account for idiosyncratic endogeneity. On the other hand, Karakaplan's (2022) approach handles idiosyncratic endogeneity through using instrumental variables but does not handle unobserved heterogeneity leaving it susceptible to exaggerating the levels of municipal cost inefficiencies. Against this background, we adopt an ad hoc approach in which the within transformation recommended by Wang and Ho (2010) is applied to all variables in the stochastic cost frontier model before ap-

plying Karakaplan's (2022) instrumental variable approach. The alternative approach of including N-1 dummy variables is less appealing as it would expose the model to the incidental parameters problem since we have a small to moderate panel dataset. The within-transformation therefore circumvented the incidental parameters problem while ensuring that Karakaplan's (2022) estimates are free from both heterogeneity endogeneity and idiosyncratic endogeneity. We particularly applied this ad hoc procedure to address the first aim as a post-estimation test found inefficiency drivers endogenous. To address the second aim of the study, we applied Greene's (2005) approach as the same test found inefficiency drivers exogenous, rendering the correction for endogeneity unnecessary.

One important aspect is noteworthy. In the Greene (2005) model, we estimate the sources of cost inefficiency through the conditional mean specification. This follows the parameterization applied in Kumbhakar et al. (1991) and Huang and Liu (1994). In the Karakaplan (2022) specification, we estimate the sources of cost inefficiency through the conditional variance specification. This parameterization involves scaling the distribution of inefficiency, as similarly applied in Caudill and Ford (1993), Caudill et al. (1995), and Hadri (1999). In both parameterizations, a negative sign on a variable implies a positive effect on cost efficiency. Similarly, a positive sign on a variable implies a negative effect on cost efficiency. The model uses lagged values of endogenous variables as instruments. The assumption is that lagged values of expenditure are predetermined and therefore exogenous to contemporaneous inefficiency shocks, making them valid instruments. Despite the plausibility of this assumption, the use of lagged instruments is not without limitations. If the error term exhibits autocorrelation, a first-order lag may still be correlated with the contemporaneous disturbance, violating the exogeneity assumption. As a robustness check, therefore, we considered in secondary regressions two external instruments, namely government grants and a 3-year rolling standard deviation of gross domestic product weighted by the population share of each municipality in the district. This reasoning is based on the notion that macroeconomic shocks and government grants, which are allocated by the national treasury, affect municipal expenditures but are exogenous to cost inefficiency.

Estimation is through the maximum likelihood estimator. It is important to mention at this stage that the panel dimensions generally dictate the type of econometric care when estimating panel data. In our case, we have a small N and small T, which makes the asymptotics of fixed N and fixed T relevant. In particular, while such dimensions downplay concerns of non-stationarity and potential slope heterogeneity across the municipalities, traditional asymptotic theory may not apply. Finite-sample issues, including bias when dealing with dynamic panel data and weak instrument problems, become severe. We do not worry about the bias, however, since our model is static and not dynamic. To mitigate finite sample problems, we bootstrap our standard errors in a bid to improve efficiency. Lastly, we perform weak instrumental variable tests to ensure their validity.

Figure 3 summarizes the methodology. The first choice is about the approach to measuring inefficiency. From the two broad approaches (the SFA and the DEA), this study prefers the SFA. Within the SFA, functional form plays a critical role in shaping the estimates of efficiency. Two main functional forms used are the Cobb–Douglas specification and the Translog. The former is relatively simple and easy to estimate but comes with restrictive assumptions of unitary elasticity of substitution and constant returns to scale. In contrast, the Translog functional form provides a second-order approximation to any twice-differentiable production function, which consequently allows for variable returns to scale and non-constant elasticities of substitution between inputs. This added flexibility makes the Translog particularly appealing in settings where input interactions are complex, albeit at the expense of higher data requirements and potential multicollinearity. The

selection between these functional forms is generally based on statistical tests. In this study, as the diagnostic tests will later show, the Cobb–Douglas specification was preferred. Within the Cobb–Douglas specification, one has to decide between estimating a dual specification (typically a cost frontier) or a primal one (a production function). In this study, both approaches are used. The dual approach, which measures efficiency from a cost perspective, is applied to the first objective, which measures municipal cost inefficiencies. The primal approach is used in the second objective, as municipalities are viewed as producers who combine inputs to provide service delivery.

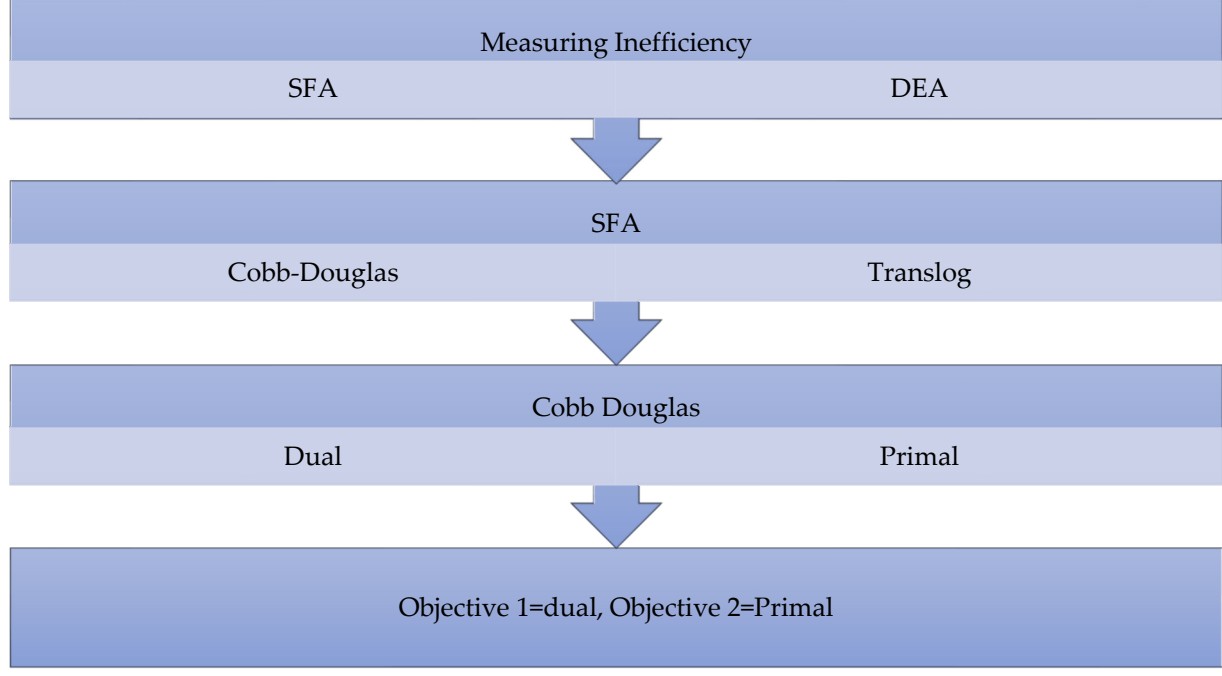

**Figure 3.** Methodological summary. Source: Own computation.

*3.7. Diagnostic Checks*

As a customary practice in empirical modelling, we conduct several diagnostic tests within the stochastic cost frontier framework to determine the reliability of the results. First, as indicated above, we assess the appropriate functional form by testing whether the Translog specification provides a better fit than the Cobb–Douglas alternative using the likelihood ratio (LR) test. The LR test statistic is calculated as LR = −2(LR(Ho)-LR(Ha)) where LR(Ho) is the LR value from the restricted Cobb–Douglas specification while LR(Ha) is the LR value from the unrestricted Translog specification. This diagnostic test is crucial, as an incorrect functional form may lead to biased efficiency estimates. Rejection of the null will be evidence in favour of the Translog specification and vice versa. Second, we examine the presence of frontier shifts over time by performing a joint significance test on time dummies. Additionally, and most importantly, we test for cost inefficiency, as the absence of cost inefficiency would render the stochastic cost frontier model unnecessary. Put differently, in the absence of cost inefficiency, the stochastic cost frontier model reduces to a standard total cost function with normal errors. In this test, LR(Ho) is the LR value from the standard ordinary least squares (OLS) regression with normal errors estimated through the glm command. LR(Ha) is the LR value from the estimated stochastic frontier model. The 5% critical values are from a mixed distribution tabulated in Kodde and Palm (1986) with one degree of freedom. A significant LR value will serve as evidence of inefficiencies justifying the stochastic frontier analysis over the standard OLS regression with normal errors. We also conduct the weak instrumental variable test recommended by Karakaplan

(2022). Lastly, since we are using the maximum likelihood estimator, we pay attention to the iteration procedure to identify any potential estimation challenge that may produce invalid results. We particularly check for the "not concave" message in the last step, as it could signal the problem of near collinearities, which may not be detected automatically by Stata, and the optimizer entering a flat region of the likelihood, which may lead to a premature declaration of convergence. We additionally check for the "backed up" message at the last step, which may arise when the optimizer works itself into a bad concave spot where the computed gradient and Hessian give a bad direction for stepping. The analysis is performed using Stata 17. The Karakaplan (2022) model is implemented using the xtsfkk command. Prior to its implementation, frontier variables are subjected to the within-transformation in Stata 17 using the command: egen M_mean = mean(M), by(id), where M is the frontier variable and id is the unique identifier variable that distinguishes the four local municipalities. This is followed by the command: gen M_within = M − M_mean. Regarding the true-fixed effects and the Battese and Coelli (1995) estimated as a robustness check with bootstrapped standard errors, we use the sfpanel command in Stata 17 with the vce(bootstrap) option. The former uses the tfe option while the latter contains bc95. The weak instruments test is implemented using the commands "est res ModelEN" after estimation. This is then followed by test iv1, iv2, . . ., ivn, where ivn signifies the n number of instruments used in the estimation. A chi2 statistic above 10 is taken as evidence of valid instruments (Karakaplan, 2022). Lastly, frontier shifts were tested through a Wald test for joint significance of time dummies. This was achieved through the Stata command testparm i.year post-estimation.

## 4. Results

Table 3 presents summary statistics. The within-transformed log values of total expenditure, water, and refuse collection all have a mean of zero, which is expected due to the within transformation. Log total expenditure exhibits the highest dispersion, followed by log water and log refuse collection. Among the included expenditure shares, employee remuneration constitutes the largest portion, followed by debt impairment and contracted services. Bulk purchases account for an average of 18.1% of total expenditure, while operating costs are 4.2%. The standard deviations of debt impairment and employee remuneration indicate moderate variability across the four municipalities. The minimum and maximum values, on the other hand, indicate considerable variation, particularly in debt impairment (ranging from 0 to 40.9%) and bulk purchases (10.7% to 27.8%). The minimum values of zero for operating costs and debt impairment are, respectively, picking up a year in the sample in which operating costs were covered by external grants and a year in which provisions for bad debts in previous years might have eliminated the need to record new impairments. A takeaway result from these summary statistics is that the dataset does not appear to be plagued by atypical observations, looking at the mean values and the range.

Table 4 presents a correlation matrix whose primary purpose is to identify any potential multicollinearity among the regressors. The presence of high collinearity makes it difficult to isolate the partial effects of regressors on both total cost and cost inefficiency. Looking at Table 4, we notice a strong positive association between log water and log refuse collection, suggesting that municipalities with higher access to water also allocate resources to waste management. Although this high collinearity could pose challenges with estimation in the cost frontier specification, we resist the temptation to drop one of the variables to prevent a potential omitted variable bias, given the strong theoretical justification for having both output variables in a cost frontier specification. The high collinearity is theoretically justified by the complementary nature of these services. Both are essential public utilities that share

infrastructure, budget allocations, and policy frameworks. Additionally, urbanization and population growth drive simultaneous demand for both services, while regulatory mandates require municipalities to manage water supply and sanitation, including waste collection, collectively. Against this background, we preferred the do-nothing approach, as our parameters remain consistent and unbiased even in the presence of multicollinearity.

**Table 3.** Descriptive Statistics.

| Variable | Obs | Mean | Std. Dev. | Min | Max |
|---|---|---|---|---|---|
| logtotalexp within | 68 | 0 | 0.632 | −1.366 | 0.89 |
| logwater within | 68 | 0 | 0.18 | −0.419 | 0.405 |
| logrefuse collection within | 68 | 0 | 0.06 | −0.126 | 0.118 |
| Bulk share (%) | 68 | 0.181 | 0.044 | 0.107 | 0.278 |
| Operating costs share (%) | 68 | 0.042 | 0.035 | 0 | 0.215 |
| Contracted services share (%) | 68 | 0.054 | 0.03 | 0.014 | 0.185 |
| Debt impairment share (%) | 68 | 0.108 | 0.091 | 0 | 0.409 |
| Employee remuneration share (%) | 68 | 0.238 | 0.053 | 0.12 | 0.337 |

**Table 4.** Correlation Matrix.

| Variables | (1) | (2) | (3) | (4) | (5) | (6) | (7) | (8) |
|---|---|---|---|---|---|---|---|---|
| (1) logtotalexp within | 1.000 | | | | | | | |
| (2) logwater within | 0.858 | 1.000 | | | | | | |
| (3) logrefuse collection within | 0.909 | 0.957 | 1.000 | | | | | |
| (4) Bulk share (%) | 0.105 | 0.049 | 0.172 | 1.000 | | | | |
| (5) Operating costs share (%) | 0.163 | 0.164 | 0.216 | 0.049 | 1.000 | | | |
| (6) Contracted services share (%) | −0.053 | −0.030 | 0.023 | 0.072 | −0.100 | 1.000 | | |
| (7) Debt impairment share (%) | 0.483 | 0.462 | 0.418 | −0.120 | −0.157 | −0.291 | 1.000 | |
| (8) Employee remuneration share (%) | −0.480 | −0.225 | −0.270 | 0.070 | 0.079 | 0.097 | −0.368 | 1.000 |

With respect to the drivers of cost inefficiency where theory is not particularly clear on variable selection, we computed variance inflation factors. As Appendix A shows, none of the potential sources of cost inefficiency have a variance inflation factor above 5, which downplays concerns of near multicollinearity. Table 5 presents the regression results. The table contains two columns of results. The first column is essentially a within-transformed version of Karakaplan (2022), which uses two external instruments, namely government grants and a 3-year rolling standard deviation of gross domestic product weighted by the population share of each municipality in the district. The second column uses internal instruments, namely lagged values of the endogenous inefficiency regressors. The upper part of the table contains the cost frontier estimates, while the lower part identifies the sources of cost inefficiency. The mean cost efficiency score in the baseline specification is 0.853, suggesting that the total cost of running a municipality was, on average, above its minimum cost level by 17.23% (i.e., $[1/(0,853)] − 1) \times 100 = 17.23\%$). Put differently, we find wasteful expenditure of 17.23%. This finding is consistent with the broader literature on public sector inefficiencies in developing countries, where governance challenges often impede optimal resource utilization (Afonso & Fernandes, 2008; Geys & Moesen, 2009; Adedeji Amusa & Fadiran, 2024; Rella et al., 2025). Looking at the results from the lower part of the table, we notice that wasteful expenditure was primarily driven by operating costs and contracted services. It is these two variables whose coefficients are positive and statistically significant at 1% and 5%, respectively. The former result is in line with Dollery and Grant (2011), who argue that outsourcing municipal services, when improperly

regulated, tends to inflate costs. The latter result agrees with Geys and Moesen (2009), who find non-productive operational spending contributing to inefficiency in local government.

**Table 5.** Municipal Cost Efficiency and its Drivers.

| Variables | External Instruments | Internal Instruments |
|---|---|---|
| Dependent variable: logtotalexpenditure | | |
| logwater (within-transformed) | 0.011 (0.536) | 0.639 (0.632) |
| Logrefusecollecton (within-transformed) | 0.885 *** (0.161) | 0.685 *** (0.193) |
| Dependent variable : $\ln\left(\sigma_u^2\right)$ | | |
| Constant | 0.812 ** (0.241) | 0.712 *** (0.236) |
| Share of bulk purchases (%) | −0.192 (0.801) | −0.674 (0.850) |
| Share of operating costs (%) | 3.157 *** (0.231) | 1.93 *** (0.646) |
| Share of contracted services (%) | 0.695 *** (0.106) | 2.256 ** (1.22) |
| Share of debt impairment (%) | 0.212 (0.232) | 0.261 (0.305) |
| Share of employee remuneration (%) | −4.788 *** (1.004) | −5.005 *** (0.878) |
| Dependent variable : $\ln\left(\sigma_w^2\right)$ | | |
| Constant | −3.322 *** (0.185) | −3.605 *** (0.215) |
| Share of bulk purchases (%) | 1.361 (1.133) | 1.212 (1.175) |
| Share of operating costs (%) | −0.782 *** (0.151) | −2.919 * (1.451) |
| Share of contracted services (%) | −1.83 *** (0.338) | −2.946 * (1.378) |
| Share of debt impairment (%) | 0.257 (0.436) | 0.264 (0.490) |
| Share of employee remuneration (%) | 0.135 (1.514) | 0.762 (1.233) |
| eta Endogeneity Test | $X^2 = 51.54$ *** $p = 0.0000$ | $X^2 = 14.15$ ** $p = 0.015$ |
| Observations | 64 | 64 |
| Log Likelihood | 10.91 | 647.70 |
| Mean Cost Efficiency | 0.867 | 0.853 |
| Median Cost Efficiency | 0.883 | 0.898 |

*, **, *** denote significance at 10%, 5%, and 1%, respectively. Figures in parentheses are standard errors. Note that a positive and significant coefficient in the cost inefficiency specification implies a negative effect on cost efficiency. The observations are 64 instead of 68 due to the use of lagged values as instruments.

The remuneration of employees has a positive and sizeable causal effect on municipal cost efficiency, which is significant at the 1% level. The positive and significant coefficient suggests that municipalities that compensate their employees well tend to operate closer to their cost frontier and therefore record less wasteful expenditure. This result is consistent with the efficiency wage hypothesis, which posits that higher wages can enhance worker productivity (Akerlof & Yellen, 1986). In the context of local municipalities and governance, this result possibly suggests that well-remunerated employees may be more motivated and accountable, which reduces the likelihood of service delays, corruption and other factors that are often associated with inefficiency in municipal operations. In addition, the result is plausible, as higher remuneration may strengthen institutional capacity and promote the efficient use of public resources. Within the literature, this finding agrees with Ncube and Monnakgotla (2016), who find adequate compensation critical for the effective functioning of local government in South Africa.

Interestingly, while debt impairment is often cited as a financial management concern for local municipalities, its coefficient is statistically insignificant in both specifications. The insignificance of debt impairment suggests that unpaid debts by households are not a significant driver of cost inefficiency of local municipalities in the district. There are two possible reasons for this result. The first possible explanation is that cost inefficiency of local municipalities may be more strongly linked to expenditure patterns rather than revenue shortfalls, as similarly argued in Afonso and Fernandes (2008). This is plausible since debt impairment in South Africa is generally more indicative of socio-economic

conditions such as poverty and unemployment than managerial inefficiency. The second possible reason is that since municipalities often adjust their budgets to accommodate non-payment, the cost structure observed in the data may have already internalized these losses, which consequently limits their marginal impact on cost inefficiency.

Lastly, the results reveal that bulk purchases exhibit a negative but statistically insignificant effect on municipal cost inefficiency. Despite being statistically significant, the negative sign is encouraging, as it shows the potential of bulk purchases to reduce municipal cost inefficiencies, likely due to economies of scale of bulk procurement. In the South African municipal context, bulk purchases, particularly of water and electricity, are often governed by long-term supply contracts, which can potentially lower average costs.

The endogeneity test returns a significant probability value, justifying the correction of endogeneity in the model. This result particularly supports the use of Karakaplan's (2022) instrumental variable approach compared to Greene's (2005) true-fixed-effects stochastic cost frontier model, which ignores idiosyncratic endogeneity.

While the above results are plausible, we consider the potential criticism that conventional asymptotic standard errors can be, in small-sample settings, unreliable due to downward bias and over-rejection of null hypotheses (Cameron & Trivedi, 2005). With only 68 observations, these concerns can be quite severe and non-trivial. Against this background, we consider additional regressions for robustness purposes in which we lag all variables to mitigate endogeneity within Greene's (2005) and Battese and Coelli's (1995) stochastic cost frontier framework and then bootstrap standard errors. Bootstrapping relies on resampling rather than large-sample theory, making it better at approximating the finite-sample distribution of estimators (Tibshirani & Efron, 1993). This approach is additionally useful in the presence of heteroskedasticity, autocorrelation, or non-normal errors (Horowitz, 2001), making it crucial to improving the reliability of inference in small samples (Davidson & MacKinnon, 2006). Table 6 presents these results. In this table, TFE Bootstrap SE represents results from the true-fixed effects of Greene (2005), while BC95 Bootstrap SE contains results from the Battese and Coelli (1995) specification. While these two secondary regressions with bootstrapped standard errors have notable limitations with respect to handling idiosyncratic endogeneity and unobserved heterogeneity, respectively, it is reassuring and comforting to note how remarkably similar their results are to the baseline model which handles both idiosyncratic endogeneity and unobserved heterogeneity. Both specifications particularly confirm operating costs and contracted services as the significant drivers of cost inefficiencies. In addition, the evidence still demonstrates the negative association between employee remuneration and cost inefficiency. Overall, therefore, our baseline results are not severely impacted by micronumerosity (problems associated with small sample sizes).

The lambda term is greater than one and significant at the 1% level in both specifications, indicating that cost inefficiencies dominate the error term. In other words, much of the variation in cost inefficiencies of these municipalities from the minimum cost level is largely a result of man-made cost inefficiencies rather than random factors beyond the control of municipal managers. This result is important, as it serves as crude evidence in support of a stochastic cost frontier model over the standard total cost regression with normal errors.

**Table 6.** Municipal Cost Efficiency and its Drivers with Bootstrapping.

| Variables | (1) TFE Bootstrap SE | (2) BC95 Bootstrap SE |
|---|---|---|
| Frontier | | |
| logwater (−1) | 0.548 *** | 0.999 *** |
| | (0.134) | (0.156) |
| Logrefuse (−1) | 0.934 *** | 0.746 *** |
| | (0.199) | (0.232) |
| Dependent variable: mu | | |
| Share of bulk purchases (%) (−1) | 1.357 | 2.406 |
| | (0.914) | (1.746) |
| Share of operating costs (%) (−1) | 0.650 *** | 1.335 *** |
| | (0.019) | (0.031) |
| Share of contracted services (%) (−1) | 1.057 *** | 0.581 *** |
| | (0.009) | (0.044) |
| Share of debt impairment (%) (−1) | 0.384 | 0.115 |
| | (0.312) | (0.432) |
| Share of employee remuneration (%) (−1) | −5.890 *** | −5.532 *** |
| | (0.578) | (0.649) |
| Constant | 14.91 *** | 14.19 *** |
| | (0.897) | (1.083) |
| Usigma | | |
| Constant | −3.0808 *** | −3.928 *** |
| | (1.173) | (1.026) |
| Vsigma | | |
| Constant | −4.042 *** | −4.381 *** |
| | (0.197) | (0.153) |
| Sigma_u | 0.214 * | 0.135 *** |
| | (0.125) | (0.014) |
| Sigma_v | 0.132 *** | 0.0636 *** |
| | (0.013) | (0.011) |
| Lambda | 1.616 *** | 2.137 *** |
| | (0.130) | (0.161) |
| Observations | 64 | 64 |
| Number of municipalities | 4 | 4 |
| Log likelihood | 62.632 *** | 75.263 *** |

Bootstrapped standard errors in parentheses. *** $p < 0.01$, * $p < 0.1$.

To achieve the second objective, we estimated a stochastic production function in which water provision and refuse collection were the output variables. We assumed that municipalities thrive to maximize the provision of water and refuse collection from the given revenue and municipal workers. In the inefficiency specification, we include cost inefficiency scores measured earlier as the regressor of interest. In this specification, population, functional literacy (the percentage of people with at least secondary school education), gross value added capturing local economic performance, and unemployment are included as controls. Noteworthy is that the endogeneity test turned out to be insignificant, rendering the instrumental variable approach unnecessary for this objective. We therefore used the true-fixed effects approach with bootstrapped standard errors. Table 7 presents these results. As the lower part of Table 7 confirms, the average efficiency score is 0.796 for water provision and 0.766 for refuse collection. This suggests that, on average, water provision and refuse collection were about 23% and 20% lower than their potential level of service provision. These scores particularly suggest that, on average, a typical municipality in Frances Baard District could have improved service provision by approximately 20–23% without requiring additional resources.

**Table 7.** Municipal Cost Efficiency and Service Delivery with Bootstrapping.

| Variables | (1) (logwater) TFE Bootstrap SE | (2) (logrefusecollection) TFE Bootstrap SE |
|---|---|---|
| Frontier | | |
| logrevenue (−1) | 0.272 *** | 0.258 *** |
| | (0.036) | (0.058) |
| logworkers (−1) | 0.635 *** | 0.768 *** |
| | (0.067) | (0.107) |
| Dependent variable: mu | | |
| Municipal cost inefficiency (−1) | 11.474 *** | 4.341 *** |
| | (0.373) | (0.319) |
| Logpopulation (−1) | 0.035 *** | 0.047 *** |
| | (0.002) | (0.001) |
| Functional literacy (−1) | −3.160 *** | −4.183 *** |
| | (0.013) | (0.013) |
| logGVA (−1) | −1.627 *** | −1.821 *** |
| | (0.301) | (0.136) |
| Unemployment (%) (−1) | 4.201 | 3.025 |
| | (3.380) | (2.835) |
| Constant | 1.935 *** | 1.265 *** |
| | (0.014) | (0.013) |
| Usigma | | |
| Constant | −3.143 | −1.336 |
| | (3.426) | (1.930) |
| Vsigma | | |
| Constant | −3.937 *** | −2.812 *** |
| | (0.187) | (0.120) |
| Sigma_u | 0.207 ** | 0.185 *** |
| | (0.110) | (0.002) |
| Sigma_v | 0.139 *** | 0.118 *** |
| | (0.013) | (0.011) |
| Lambda | 1.487 *** | 1.568 *** |
| | (0.356) | (0.254) |
| Observations | 64 | 64 |
| Number of municipalities | 4 | 4 |
| Log likelihood | 32.72 *** | 45.26 *** |
| Mean efficiency | 0.796 | 0.766 |

Bootstrapped standard errors in parentheses. *** $p < 0.01$, ** $p < 0.05$.

With respect to the main explanatory variable, we observe a positive and significant coefficient of municipal cost inefficiency across both specifications. The positive sign indicates that higher cost inefficiency of municipalities increases the gap between actual and potential service provision. This result validates the proposition that municipal cost inefficiencies contribute to poor service delivery. Since the model is non-linear, the coefficient on cost inefficiency cannot be interpreted as a marginal effect. To compute the marginal effect, we used the predict marginal command following Kumbhakar et al. (2015) post-estimation. The computed marginal effects of cost inefficiency are presented in Table 8. A percentage point increase in municipal cost inefficiency is associated with a 0.096% increase in the service gap for water provision and a 0.0085% increase in the service gap for refuse collection. These elasticities underscore the detrimental impact of municipal cost inefficiency on service delivery.

**Table 8.** Marginal effects of Municipal Cost Efficiency.

| Variables | (1) (Logwater) TFE Bootstrap SE | (2) (Logrefusecollection) TFE Bootstrap SE |
|---|---|---|
| Mean | 0.096 | 0.008 |

From Table 7, functional literacy enters with a negative and significant coefficient, as expected. This result indicates that a literate population is better equipped to demand accountability and participate in public debates for better governance (Glaeser et al., 2004). In the context of municipal service delivery, higher functional literacy may also improve the responsiveness and administrative capacity of local governments, which contributes to more efficient service delivery. The negative effect of gross value added (GVA) needs to be interpreted with caution. While it may be tempting to argue that higher GVA reflects stronger local economic activity, which can enhance a municipality's revenue base and improve service delivery following Bahl et al. (1992), this explanation does not apply here, as the frontier specification is holding constant each municipality's total revenue. A more plausible explanation is that municipalities with higher GVA tend to have better infrastructure, better connectivity, and stronger institutional frameworks, all of which facilitate better coordination and delivery of services such as water provision and waste management (Rodríguez-Pose & Gill, 2004).

The positive coefficient of population indicates that larger municipal populations are associated with poor service delivery. This finding may be attributed to the increased demand normally placed on municipalities as populations grow, which can strain infrastructure and institutional capacity, particularly in contexts with limited planning and constrained resources. With regard to unemployment, we do not find its effect on service delivery statistically significant. Since unemployment tends to increase with population growth, the significance could be reflecting the fact that its effect on service delivery is indistinguishable from that of population growth.

*Diagnostic Tests*

Table 9 presents the results from diagnostic tests performed to assess the reliability of the regression results. As the results show, the likelihood ratio is insignificant in our functional form tests. This indicates that the restricted model (the Cobb-Douglas specification) is preferable over the unrestricted model (the Translog specification). This result is true for both objectives 1 and 2. Evidence further shows that null hypotheses of no inefficiencies are strongly rejected by the LR tests, indicating that the estimated stochastic frontier models were more appropriate compared to standard regressions with normal errors. Lastly, the null hypothesis of weak instrument is strongly rejected, suggesting that the instruments used in the baseline specification of the first objective are valid. The endogeneity test for the second objective returns an insignificant probability value, rendering the correction for endogeneity unnecessary for this objective. Lastly, the Hausman specification tests favored the fixed effects transformation over the inclusion of random effects, while the Wald test for joint significance found time dummies statistically insignificant at the 10% level in both specifications. These diagnostic tests, therefore, support the within-transformation in all specifications without time dummies.

**Table 9.** Diagnostic Tests.

| Test | Test Statistic | Implication |
|---|---|---|
| Functional form (first objective) | LR stat = 1.03 | Cobb-Douglas specification |
| Functional form (second objective) | LR stat = 1.12 | Cobb-Douglas specification |
| Cost inefficiencies | LR stat = 135.14 *** | Stochastic cost frontier model |
| Service delivery inefficiencies | LR stat = 102.42 *** | Stochastic production frontier model |
| Weak instrumental variable test | $Chi^2(25)$ = 295.65 *** | Valid instruments |
| Endogeneity test (second objective) | $X^2$ = 2.12, $p$ = 0.347 | Exogeneity |
| Fixed vs random effects (first objective) | $p$ = 0.031 | Fixed effects (within-transformation) |
| Fixed vs random effects (second objective) | $p$ = 0.042 | Fixed effects (within-transformation) |
| Time dummies (first objective) | $F(15, 46)$ = 1.17, $p$ = 0.3291 | No frontier shifts |
| Time dummies (second objective) | $F(15, 46)$ = 1.21, $p$ = 0.2974 | No frontier shifts |

*** denotes significance at the 1% level.

## 5. Discussion

The study has addressed two objectives. The first sought to measure the degree of cost inefficiency and estimate its key sources. The second sought to identify how the measured municipal cost inefficiencies affect service delivery with a specific focus on water provision and refuse collection. The findings from the first objective indicate that municipalities in South Africa are operating with a considerable degree of inefficiency. The result particularly observed that local governments are, on average, spending over 17% more than the minimum cost required to deliver a given level of service delivery. This result echoes prior studies which document inefficiency in public sector operations (Afonso & Fernandes, 2008; Geys & Moesen, 2009).

From a policy and governance perspective, this level of inefficiency aligns with the recurring findings of the Auditor-General of South Africa, whose reports have consistently raised concerns over irregular expenditure, non-compliance with procurement regulations, and inadequate financial oversight within municipalities. A look into the sources of municipal cost inefficiencies finds operating expenditures and contracted services the most influential drivers. Both variables are positively and significantly associated with inefficiency, suggesting that these spending categories are not being effectively controlled and optimally allocated in these municipalities. This is in line with earlier findings by Geys and Moesen (2009), who found administrative bloat and ineffective spending practices associated with high inefficiency in the local public sector. In South Africa, and looking at our metadata, this finding is not surprising since operating costs include general expenses, travel allowances, entertainment and support functions that may not directly enhance service provision. Contracting services, although potentially beneficial in theory, have often been associated with limited transparency, rent-seeking activities, and limited accountability (Dollery & Grant, 2011). This result particularly corroborates the view that municipalities that outsource core functions may record higher costs and poor value for money in the absence of transparency and accountability.

In the second objective, we found that municipal cost inefficiency increases the gap between actual and potential service delivery, which is consistent with the empirical literature on public sector efficiency. Several studies have documented a negative relationship between cost inefficiency and the quality or quantity of public services. For instance, Worthington and Dollery (2001) found that Australian local governments with higher inefficiency scores tended to underperform in service delivery, especially in utilities and community services. Similarly, studies in developing country contexts, such as that by Coelli (2003), show that inefficiencies in municipal resource allocation led to poor service

delivery, particularly in essential services such as water, sanitation, and waste management. The positive association observed in this study supports these findings by confirming that inefficiency not only results in wasteful expenditure but also directly impairs service outcomes. In addition, the results agree with De Borger and Kerstens (1996), who argue that inefficiencies in local governments are often driven by weak fiscal discipline, which ultimately contributes to poor service delivery.

## 6. Limitations of the Study

This study is subject to two main limitations that should be considered when interpreting the findings. First, the analysis is based on data from only four municipalities within the Frances Baard District in the Northern Cape, resulting in a small sample size of 64 observations (four municipalities over 16 years). While bootstrapping methods were applied to mitigate potential efficiency losses arising from the small sample, the limited cross-sectional dimension may constrain the robustness of the findings and limit the generalizability of the results to municipalities in other regions with different structural and economic characteristics.

Second, the study focuses exclusively on municipalities in a desert economy characterized by a high dependency on diamond mining, cyclical revenue patterns, and limited economic diversification. The unique economic and governance dynamics of the Frances Baard District imply that the findings may be more relevant to municipalities operating in similar arid, resource-dependent contexts. Thus, caution should be exercised in generalizing the results to municipalities in other settings, particularly those with more diversified economies or different climatic conditions.

## 7. Conclusions

This study sought to examine the extent and sources of cost inefficiency in South African municipalities and to assess the implications of these inefficiencies for service delivery, focusing on the local municipalities in the Frances Baard District of the Northern Cape. The results reveal that municipalities operate with inefficiencies, spending on average 17.23% more than the minimum cost required to deliver existing services. This confirms persistent inefficiencies in local government operations, which is consistent with earlier empirical studies and ongoing concerns raised by the Auditor General. Operating costs and contracted services emerge as the primary drivers of this inefficiency, highlighting areas where expenditure practices are misaligned with service delivery outcomes. These findings reflect broader governance and institutional shortcomings that weaken expenditure control and promote wasteful spending. The study also establishes a negative relationship between cost inefficiency and municipal service delivery, suggesting that inefficiency is not merely wasteful expenditure but also a significant constraint to service delivery at the local level.

Overall, the evidence underscores the urgent need for reforms aimed at strengthening expenditure oversight, improving contract management, and investing in administrative capacity. The results particularly raise an urgent need for these municipalities to cut operational spending, particularly on non-priority areas such as entertainment and travel allowances. More specifically, municipalities may consider improving procurement oversight mechanisms and performance-based budgeting. Implementing these measures to address cost inefficiencies is critical to enhancing the value for money in municipal spending and improving the quality of life for communities reliant on public services. Future studies may consider an examination of the institutional and governance factors that potentially mediate the relationship between municipal cost inefficiency and service delivery outcomes across municipalities.

**Funding:** This research received no external funding.

**Institutional Review Board Statement:** Not applicable.

**Informed Consent Statement:** Not applicable.

**Data Availability Statement:** The raw data supporting the conclusions of this article will be made available by the authors on request.

**Conflicts of Interest:** The author declares no conflicts of interest.

## Appendix A

**Table A1.** VIF Table—Cost Inefficiency Drivers.

|                        | VIF   | 1/VIF |
|------------------------|-------|-------|
| Debt impairment        | 1.445 | 0.692 |
| Operating costs        | 1.205 | 0.83  |
| Contracted services    | 1.197 | 0.836 |
| Employee remuneration  | 1.161 | 0.861 |
| Bulk purchases         | 1.052 | 0.95  |
| Mean VIF               | 1.212 | .     |

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
