# Peer review of "Modelling Municipal Cost Inefficiencies in the Frances Baard District of South Africa and Their Impact on Service Delivery"

_admsci, doi:10.3390/admsci15060229_

Round 1

Reviewer 1 Report

Comments and Suggestions for Authors

Dear authors,

Your research paper addresses cost inefficiency and service delivery in South African municipalities using a stochastic frontier analysis (SFA) approach on a micro-panel dataset. The topic is relevant, and the methods used are sophisticated and technically appropriate. However, despite the paper's good intentions and structure, there are major issues that must be addressed by the authors

Your study uses advanced econometric techniques and demonstrates a good grasp of methodological challenges in panel data. The diagnostic testing and robustness checks are strong points.

However, the paper lacks a deeper conceptual framework. It jumps too quickly into empirical modeling without thoroughly developing the theory of cost inefficiency in local government (e.g., principal-agent problems, public choice theory, fiscal federalism). Therefore I recommend the authors to add a full theoretical section explaining why inefficiency happens in municipalities, using broader economic or political economy frameworks.

The notations for the cost and production frontiers are cluttered, repetitive, and sometimes confusing.Equations are poorly explained so I suggest the authors to simplify and clarify model specification. Add a diagram summarizing the modeling strategy if possible.

When it comes to the writing style, several sentences are overly long, complex, and redundant. For example, the description of the estimation strategy is unnecessarily complicated. Some minor grammatical errors and typos are present ( “preferrable” instead of “preferable”, "am" instead of "an" on p.5 ("cost frontier variable in the first objective and am Output in the second", "expenditure practices are either poorly managed or misaligned with service delivery outcomes" - slightly repetitive).

The conclusion to "cut operational costs" and "enhance accountability" is obvious. More specific, actionable policy recommendations are needed like improving procurement oversight mechanisms, performance-based budgeting, public participation frameworks….

In Table 2, the correlation matrix is discussed, but no Variance Inflation Factors (VIF) are provided to formally assess multicollinearity. In the methods section, some technical steps (such as the within-transformation) are repeated excessively and could be streamlined.

A few references (e.g., Adedeji Amusa & Fadiran 2024) are cited multiple times , you need to consider diversifying citations.

Table 8 caption says "Marginal effects of Municipal Cost Efficiency," but the actual table is about Diagnostic Tests.

Finally, the article shows strong potential but must undergo substantial revisions to address conceptual framing, generalizability, clarity of methodology and contextual relevance.

Good Luck

Author Response

Reviewer One

Your research paper addresses cost inefficiency and service delivery in South African municipalities using a stochastic frontier analysis (SFA) approach on a micro-panel dataset. The topic is relevant, and the methods used are sophisticated and technically appropriate. However, despite the paper's good intentions and structure, there are major issues that must be addressed by the authors.

Your study uses advanced econometric techniques and demonstrates a good grasp of methodological challenges in panel data. The diagnostic testing and robustness checks are strong points.

However, the paper lacks a deeper conceptual framework. It jumps too quickly into empirical modeling without thoroughly developing the theory of cost inefficiency in local government (e.g., principal-agent problems, public choice theory, fiscal federalism). Therefore, I recommend the authors to add a full theoretical section explaining why inefficiency happens in municipalities, using broader economic or political economy frameworks.

Thank you for the comment. My prior omission of the theoretical section was based on the journal’s template which has Introduction followed by Materials and Methods. I however agree with your suggestion, and I have added not only a full theoretical section but also an empirical literature review section.

The notations for the cost and production frontiers are cluttered, repetitive, and sometimes confusing. Equations are poorly explained so I suggest the authors to simplify and clarify model specification. Add a diagram summarizing the modeling strategy if possible.

Thank you for raising this. Kindly check the new model specifications in the methodology section highlighted in yellow. I have systematically and separately specified the models for objectives one and two. I hope the new specifications provide clarity that is sufficient enough to eliminate the need for a diagram.

When it comes to the writing style, several sentences are overly long, complex, and redundant. For example, the description of the estimation strategy is unnecessarily complicated. Some minor grammatical errors and typos are present ( “preferrable” instead of “preferable”, "am" instead of "an" on p.5 ("cost frontier variable in the first objective and am Output in the second", "expenditure practices are either poorly managed or misaligned with service delivery outcomes" - slightly repetitive).

I have visited the specific sections and corrected accordingly. I am hoping that any remaining errors about writing style will be addressed during the editorial process.

The conclusion to "cut operational costs" and "enhance accountability" is obvious. More specific, actionable policy recommendations are needed like improving procurement oversight mechanisms, performance-based budgeting, public participation frameworks….

Thank you for your suggestion. I have included your specific suggestions in the conclusion section.

In Table 2, the correlation matrix is discussed, but no Variance Inflation Factors (VIF) are provided to formally assess multicollinearity. In the methods section, some technical steps (such as the within-transformation) are repeated excessively and could be streamlined.

Thank you for raising this. However, I respectfully do not see the need for VIFs considering they serve the same purpose as a correlation matrix.

A few references (e.g., Adedeji Amusa & Fadiran 2024) are cited multiple times, you need to consider diversifying citations.

Thank you raising this suggestion. As I indicated earlier, the new draft has added a full section of empirical literature which I hope addresses this concern.

Table 8 caption says "Marginal effects of Municipal Cost Efficiency," but the actual table is about Diagnostic Tests.

This has been corrected. Thank you.

Finally, the article shows strong potential but must undergo substantial revisions to address conceptual framing, generalizability, clarity of methodology and contextual relevance.

Thank you

Reviewer 2 Report

Comments and Suggestions for Authors

General comments

This study addresses the relationship between cost inefficiency and service delivery in four municipalities in the Frances Baard District of South Africa, using stochastic frontier analysis (SFA).

This is an interesting and novel topic for public finance. The manuscript is well organized and written, demonstrating technical rigor, but suffers from some design and presentation limitations.

In my opinion, the most important limitation of this research is that the findings are concentrated in only four municipalities in the same district, which limits the extrapolation of the results. I believe the article should more explicitly acknowledge this external constraint and justify why, despite the narrow scope, the results provide transferable lessons.

There are different types of efficiency. The article only addresses cost efficiency. I think this choice needs to be justified a bit.

The time frame (2006-2023) is extensive, but the annual aggregation may obscure significant shocks, such as electoral or regulatory changes. Discussion of this selection would help gauge potential biases.

Two indicators are used: water and waste. The portfolio of municipal services is much broader. The estimated boundary may not be representative of other municipalities with highly diversified services. This would need to be justified.

Are the cost series in nominal or constant units? Inflation over 18 years can distort the border if it is locked in nominal units.

You've opted for the Cobb-Douglas function. This assumes constant returns to scale and unitary elasticities, right?
This assumption, which doesn't seem realistic to me, needs to be explained.

The effective sample is 64 observations (4 × 16), which provides very few degrees of freedom for a model with more than 12 parameters in some equations. Even though bootstrap errors are applied, the variance of the estimators may still be underestimated.

The narrative assumes that any deviation from the border is “superfluous expenditure”. Is this realistic? Cost inefficiency could reveal heterogeneity in service quality or superior service standards, right?

Cost savings have a fiscal impact: shouldn't they be analyzed?

Others

The introduction should clearly state the objectives of the research and the hypotheses raised. 

In section 2, it would be appropriate to explain in more detail what the specific knowledge gap is that this research addresses.

Line 109: What is Eskom debt? Is Eskom an electricity company?

Furthermore, the use of variables as a percentage of total expenditure can introduce multicollinearity. Given the time-based panel, it would be appropriate to run stationarity and autocorrelation tests.

I think the model specifications need to be better explained.

Table 4. Can the log Likelihood go from 10.91 to 647.70?

Improve format of tables 2 to 5.

Author Response

Reviewer Two

This study addresses the relationship between cost inefficiency and service delivery in four municipalities in the Frances Baard District of South Africa, using stochastic frontier analysis (SFA).

This is an interesting and novel topic for public finance. The manuscript is well organized and written, demonstrating technical rigor, but suffers from some design and presentation limitations.

In my opinion, the most important limitation of this research is that the findings are concentrated in only four municipalities in the same district, which limits the extrapolation of the results. I believe the article should more explicitly acknowledge this external constraint and justify why, despite the narrow scope, the results provide transferable lessons.

Thank you for raising this. I have added a section in both the introduction (as a contribution of the study) and in the methodology section justifying the exclusive focus on the four municipalities. The main selling point is that these municipalities belong to a desert economy which, as documented in literature, creates unique circumstances for local municipalities to navigate in terms of revenue collection and expenditure patterns. I have elaborated on this aspect in the revised section. Perhaps more importantly, I have added a limitation of the study section where I explicitly site the focus on four municipalities as a potential limitation. The mitigating factors in my opinion are the benefits of focusing on municipalities belonging to a desert economy and the bootstrapping of standard errors.

There are different types of efficiency. The article only addresses cost efficiency. I think this choice needs to be justified a bit.

Thank you for raising this. I have added a section justify the focus on cost efficiency. This is included in the methodology section under measurement of cost efficiency.

The time frame (2006-2023) is extensive, but the annual aggregation may obscure significant shocks, such as electoral or regulatory changes. Discussion of this selection would help gauge potential biases.

Thank you for the comment. I am of the view that the nature of such regulatory factors is taken care of by the within-transformation especially when one considers that elections normally take place in the same year across all municipalities. I therefore do not anticipate any bias arising from this.

Two indicators are used: water and waste. The portfolio of municipal services is much broader. The estimated boundary may not be representative of other municipalities with highly diversified services. This would need to be justified.

Thank you for the comment. I have added a section in the methodology that justifies the focus on water and refuse collection over other services offered by municipalities

Are the cost series in nominal or constant units? Inflation over 18 years can distort the border if it is locked in nominal units.

They are deflated using constant 2015 prices at the data source.

You've opted for the Cobb-Douglas function. This assumes constant returns to scale and unitary elasticities, right?
This assumption, which doesn't seem realistic to me, needs to be explained.

Thank you for the comment. The choice of the Cobb-Duglas is backed by diagnostic tests in Table 9.

The effective sample is 64 observations (4 × 16), which provides very few degrees of freedom for a model with more than 12 parameters in some equations. Even though bootstrap errors are applied, the variance of the estimators may still be underestimated.

This comment is appreciated. However, the raised concern is all about a trade-off between bias and efficiency. Dropping some control variables (to increases degrees of freedom) potentially invites an omitted variable bias while making it difficult to isolate the effect of cost inefficiency on service delivery. I opted to prevent endogeneity which I think is a more serious problem and mitigate efficiency losses from fewer degrees of freedom through bootstrapping.

The narrative assumes that any deviation from the border is “superfluous expenditure”. Is this realistic? Cost inefficiency could reveal heterogeneity in service quality or superior service standards, right?

This is partially true. In my opinion, however, service quality is something that is in most cases specific to each municipality and therefore less likely to reflect in efficiency scores when one uses a within-transformation.

Cost savings have a fiscal impact: shouldn't they be analyzed?

They do have a fiscal impact in certain instances but if municipalities are found to be cost inefficient as we did in this study, it follows that costs were above the minimum possible level which consequently washes away the relevance of incorporating savings into the model.

Others

The introduction should clearly state the objectives of the research and the hypotheses raised. 

I have added this in lines 51-56.

In section 2, it would be appropriate to explain in more detail what the specific knowledge gap is that this research addresses.

The gap mentioned in the introduction relates to the lack of studies in South Africa that specifically focus on cost inefficiency of municipalities.

Line 109: What is Eskom debt? Is Eskom an electricity company?

It is a parastatal that supplies electricity at national level. This is mentioned in the methodology section of the revised manuscript.

Furthermore, the use of variables as a percentage of total expenditure can introduce multicollinearity. Given the time-based panel, it would be appropriate to run stationarity and autocorrelation tests.

Differencing variables in a stochastic frontier framework distorts the measurement of cost inefficiencies. This is why stationarity tests are not popular in studies applying this method.

I think the model specifications need to be better explained.

I have re-specified the specifications in a systematic manner as alluded to in my response to reviewer 1. I hope this revision addresses your concern.

Table 4. Can the log Likelihood go from 10.91 to 647.70?

Thank you for the question. Yes, as these are two totally different models. One assumes exogeneity and therefore follows a totally different estimation process compared to the other.

Improve format of tables 2 to 5.

Thank you for your suggestion, however I would like to highlight that I followed the format of the journal.

Reviewer 3 Report

Comments and Suggestions for Authors

The study examines cost inefficiency and its impact on service delivery in four South African municipalities from 2006 to 2023. It finds that municipalities spent 23% more than necessary and delivered 23% less than their potential. Key contributors to inefficiency include high operating costs and mismanaged contracted services. The study recommends cutting non-essential expenses and improving accountability to enhance service delivery.

The number of municipalities included in the study is small, limited to only 4 units. This small sample size means that the data analysis lacks methodological robustness. Moreover, considering that the total number of municipalities is 257, the sample appears to be poorly representative.

The objectives of the study should be clearly stated in the Introduction section, not just in the abstract. Additionally, the authors should support the reader in understanding the relevance of the study by thoroughly explaining the rationale behind the chosen objectives.

The literature review should be expanded to include international references. There is a substantial body of literature that has examined municipal efficiency in the provision of public services, particularly in the areas of integrated water services and solid waste management.

The specification of models 1, 2, 3, and 4, along with the variables used in the study, should be supported by a more in-depth literature review. The authors should identify relevant variables based on similar studies. Specifically, each variable included in the model should be justified by at least one or two prior studies that have used the same variable. In cases where a variable is used that has not appeared in previous literature, the authors should provide appropriate justification for its inclusion.

I also recommend clearly distinguishing between dependent and independent variables by presenting two separate tables (instead of Table 1): one associated with the SFA model used for the first research objective and the other for the SFA model related to the second objective.

Regarding the data source (QUANTEC), I suggest it be cited only in the text. Since it is the sole source used, the column referencing it in Table 1 could be removed, and the information regarding the source could instead be placed in a footnote below the table.

The stochastic frontier model presented in the study is based on a panel consisting of only four observed units over a period of 17 years (64 observations). While the length of the panel and the implementation of bootstrapping may partially compensate for the limited cross-sectional dimension, the use of such a small number of units raises significant concerns regarding the robustness and reliability of the estimates. Specifically, the low heterogeneity among units may hinder the effective identification of the technical inefficiency term, which in this type of model is estimated through comparisons across units. Moreover, including a relatively large number of explanatory variables in a context with only four units may lead to overfitting and instability in the estimated coefficients. It is therefore recommended to consider reducing the number of regressors or, alternatively, to explore non-parametric methods (such as Data Envelopment Analysis, DEA) or Bayesian approaches, which may offer greater flexibility in the presence of limited sample sizes.

The authors should highlight the limitations of the study.
Additionally, they should explain whether the results can be generalized to other contexts (in the country or abroad).

Author Response

Reviewer Three

The study examines cost inefficiency and its impact on service delivery in four South African municipalities from 2006 to 2023. It finds that municipalities spent 23% more than necessary and delivered 23% less than their potential. Key contributors to inefficiency include high operating costs and mismanaged contracted services. The study recommends cutting non-essential expenses and improving accountability to enhance service delivery.

The number of municipalities included in the study is small, limited to only 4 units. This small sample size means that the data analysis lacks methodological robustness. Moreover, considering that the total number of municipalities is 257, the sample appears to be poorly representative.

Thank you for your comment. As you may have read from my earlier responses above, I added a section which justifies the study area. The added section essentially justifies why the study exclusively focused on the four municipalities. I have also added a section for limitations of the study where I cited this as a potential limitation. My main selling point is that the benefits of studying municipalities in such a desert economy outweigh the methodological limitation of a small sample size which I partially addressed using bootstrapped standard errors as a robustness exercise.

The objectives of the study should be clearly stated in the Introduction section, not just in the abstract. Additionally, the authors should support the reader in understanding the relevance of the study by thoroughly explaining the rationale behind the chosen objectives.

I have added the objective and hypothesis statement in the introduction (see lines 51-56). Regarding the suggestion of adding a thorough rationale, I am constrained by the number of words as the word count is already above 12000. I am of also of the view that the first and second paragraphs have done a decent job in giving the reader a rationale for the study.

The literature review should be expanded to include international references. There is a substantial body of literature that has examined municipal efficiency in the provision of public services, particularly in the areas of integrated water services and solid waste management.

Thank you for suggesting this. I have added a stand-alone empirical literature review section which includes international studies.

The specification of models 1, 2, 3, and 4, along with the variables used in the study, should be supported by a more in-depth literature review. The authors should identify relevant variables based on similar studies. Specifically, each variable included in the model should be justified by at least one or two prior studies that have used the same variable. In cases where a variable is used that has not appeared in previous literature, the authors should provide appropriate justification for its inclusion.

This is a good point. I have incorporated this. Please see the revised version under the equations. The inclusion of variables is justified by a combination of theory and empirical literature.

I also recommend clearly distinguishing between dependent and independent variables by presenting two separate tables (instead of Table 1): one associated with the SFA model used for the first research objective and the other for the SFA model related to the second objective.

I have addressed this, thank you.

Regarding the data source (QUANTEC), I suggest it be cited only in the text. Since it is the sole source used, the column referencing it in Table 1 could be removed, and the information regarding the source could instead be placed in a footnote below the table.

This has been addressed accordingly.

The stochastic frontier model presented in the study is based on a panel consisting of only four observed units over a period of 17 years (64 observations). While the length of the panel and the implementation of bootstrapping may partially compensate for the limited cross-sectional dimension, the use of such a small number of units raises significant concerns regarding the robustness and reliability of the estimates. Specifically, the low heterogeneity among units may hinder the effective identification of the technical inefficiency term, which in this type of model is estimated through comparisons across units. Moreover, including a relatively large number of explanatory variables in a context with only four units may lead to overfitting and instability in the estimated coefficients. It is therefore recommended to consider reducing the number of regressors or, alternatively, to explore non-parametric methods (such as Data Envelopment Analysis, DEA) or Bayesian approaches, which may offer greater flexibility in the presence of limited sample sizes.

I apologize for referring you to my earlier responses regarding the issue of 64 observations where I mentioned the desire to exclusively focusing on desert municipalities while mitigating small sample problems using bootstrapping methods as you correctly observed. Regarding other estimation strategies proposed, the limitations of the DEA are highlighted in the methodology section. Being deterministic makes it susceptible to overstating inefficiencies. The Bayesian SFA approach is appealing but I would suggest using it in one of my future studies in this area as it uses a totally different methodology.

The authors should highlight the limitations of the study.

I have done so, thank you.

Additionally, they should explain whether the results can be generalized to other contexts (in the country or abroad).

Thank you for raising this. I have explained this in the limitations section.

Round 2

Reviewer 1 Report

Comments and Suggestions for Authors

Dear authors,

Thank you for the recent updates, the manuscript has improved. However, I believe that some concerns weren’t taken into consideration and I believe the authors need to adress them:

1- The newly added tables are clear and useful, and you have modified the model specifications to achieve the required separation for objectives one and two.Having said that, I would nevertheless consider including a diagram (visual representation) outlining the modeling method to avoid some confusion among readers unfamiliar with the technical background. It would enhance, not replace, the text and tables!

2- While the comprehensive methodology description has improved, the correlation matrix provides an initial overview of the interactions between model variables. I recommend using (adding) Variance Inflation Factors (VIFs) to analyze multicollinearity. Unlike correlation matrices (which reflect pairwise linear relationships), VIFs indicate how much the variance of each regression coefficient is inflated due to collinearity with all other predictors. They are used in models with multiple independent variables because high multicollinearity may not be apparent through pairwise correlations alone. It would also directly address any concerns and issues about the reliability of the regression coefficients.

Good luck

Author Response

Dear Editor and Reviewers,

Thank you for the second-round review. I have carefully considered each comment and have revised the manuscript accordingly to improve its clarity, methodological rigour, and overall contribution. In what follows, I provide a point-by-point response to the reviewers’ comments. For each concern raised, I explain how I have addressed it in the revised manuscript or clarify where the issue had already been covered in the original version. I hope the revisions meet the expectations of the editorial team and the reviewers, and I remain at your disposal for any further clarification.

Reviewer One

Comment 1

Thank you for the recent updates, the manuscript has improved. However, I believe that some concerns weren’t taken into consideration, and I believe the authors need to address them:

The newly added tables are clear and useful, and you have modified the model specifications to achieve the required separation for objectives one and two. Having said that, I would nevertheless consider including a diagram (visual representation) outlining the modeling method to avoid some confusion among readers unfamiliar with the technical background. It would enhance, not replace, the text and tables!

Response

I am pleased to hear that the recent updates have improved the manuscript and that the revised tables and model specifications are now clearer. Regarding your suggestion to include a diagram outlining the modeling approach, I appreciate the recommendation and fully agree that a visual representation would enhance clarity for readers, particularly those less familiar with the technical aspects of the stochastic frontier framework. I have therefore developed and included a schematic diagram in the revised manuscript (Figure 3) which visually summarizes the methodological choices made in the study. I have also supplemented this with a narrative account to provide context. This is highlighted in yellow. I hope this is done to your satisfaction otherwise I am happy to make additional improvements.

Comment 2

While the comprehensive methodology description has improved, the correlation matrix provides an initial overview of the interactions between model variables. I recommend using (adding) Variance Inflation Factors (VIFs) to analyze multicollinearity. Unlike correlation matrices (which reflect pairwise linear relationships), VIFs indicate how much the variance of each regression coefficient is inflated due to collinearity with all other predictors. They are used in models with multiple independent variables because high multicollinearity may not be apparent through pairwise correlations alone. It would also directly address any concerns and issues about the reliability of the regression coefficients.

Response

Thank you for your constructive comment regarding the potential issue of multicollinearity and the suggestion to include a VIF table. While I acknowledge the usefulness of VIF as a diagnostic tool, I provided a theoretical discussion highlighting the complementarity between key regressors in the frontier specification, particularly log water and log refuse collection. These variables exhibit high pairwise correlation, which we attribute to shared infrastructure, overlapping service mandates, and co-movement driven by urbanization and population growth. Despite this correlation, we opted to retain both variables in the cost frontier model to avoid omitted variable bias, as their inclusion is strongly supported by the theoretical structure of municipal service delivery. Nonetheless, in response to your concern, I have now computed and included the VIFs in the appendix section primarily for the drivers of cost inefficiency. The results show that all VIF values fall below the conventional threshold of 5, confirming that multicollinearity, while present, is not at a level that undermines the reliability of the model estimates.

Reviewer 2

Comment 1

While the authors acknowledge the limitation of a small sample - comprising only four municipalities observed over 16 years - the application of SFA in this context raises substantial methodological concerns. SFA typically requires sufficient cross-sectional variation to reliably distinguish inefficiency from statistical noise. With only four cross-sectional units, there is a significant risk that inefficiency effects are not separately identifiable and estimates of efficiency scores may be highly unstable or driven by unit-specific idiosyncrasies rather than systematic differences. Moreover, SFA relies on strong distributional assumptions about the inefficiency and noise terms, which are difficult to validate or test with such a limited number of municipalities. While the use of bootstrapping to mitigate efficiency losses in estimation is noted, it does not compensate for the fundamental issue of under-identification and limited generalizability. In fact, bootstrapped standard errors in this setting may give a false sense of robustness. Given these concerns, the validity of the results, both in terms of internal consistency and external relevance, is questionable. I strongly recommend that the authors consider alternative methodological approaches more suited to small-sample analysis. These could include case study approaches or qualitative comparative analysis (QCA), which can provide richer contextual insight when sample sizes are small. Alternatively, the authors should provide a stronger justification for the use of SFA under such constraints, including sensitivity tests and robustness checks to assess the stability of their findings across model specifications.

Response

We sincerely appreciate the continued engagement on this issue. We would like to respectfully note that this issue was addressed in the originally submitted manuscript, specifically in table 6 where we acknowledged the limitations of small-sample inference and implemented appropriate mitigating measures. In particular, as you have recommended in your last paragraph, we conducted additional robustness checks using two alternative SFA specifications, Greene’s (2005) true fixed effects model and Battese and Coelli’s (1995) time-varying inefficiency model, both estimated with bootstrapped standard errors to improve inference reliability. This is discussed on lines 826 – 849 of the revised manuscript. We appreciate your last comment that such robustness checks will help assess the stability of the findings across model specifications. However, as you will note on line 842 and line 847, we particularly conclude that the results from the alternative specifications are remarkably similar and that the baseline results are not severely influenced by problems of small sample size.

I would also like to add that, while I fully acknowledge that small sample sizes can pose methodological challenges for the SFA, particularly in separating inefficiency effects from statistical noise, leading literature has shown that the method can still be useful in contexts where the limited sample is a deliberate and justified choice driven by the desire to focus on a targeted group of units with for example shared institutional, geographic, or policy characteristics. One of these scholars is Greene (2010) who, interestingly developed one of the approaches that I applied as a robustness exercise, applied an SFA model with only 30 observations in a study that sought to focus exclusively on OECD countries in a secondary regression. While the relatively small sample size rightly invites criticism of micronumerosity, I am of the view that the author considered the need to isolate a specific policy-relevant subgroup. Similarly, in this study, I am of the view that the small sample of four municipalities over 16 years (giving 68 observations) reflects a deliberate desire to focus on a subnational governance context which I have justified both in the introduction and methodology sections. I have additionally, as you rightly acknowledged, implemented measures to mitigate concerns that accompany small sample problems, including alternative model specifications and bootstrapped standard errors to ensure the robustness of our findings despite the limited number of cross-sectional units. I sincerely hope that this account addresses your concerns on micronumerosity.

Greene, W., 2010. A stochastic frontier model with correction for sample selection. Journal of productivity analysis34, pp.15-24.

Comment 2

The authors should carefully check the accuracy of the text. There are several typographical and grammatical/stylistic errors. For example, on line 286: “…Thei findings…” — the word “Thei” should be corrected by removing the letter “i”. Also, on line 290: “…the auditor general…” — do the authors mean “generally” or “in general”?

Response

Thank you for noting this. I have corrected Thei to reflect Their. With respect to auditor general, it should read the Auditor General as it is a position established in the South African constitution. The Auditor-General of South Africa is accountable to the National Assembly in terms of section 181(5) of the Constitution and section. I am hopeful that any remaining typos will be addressed by the editorial services of the journal and invoiced accordingly.

Reviewer 3

Comment

In my opinion, the changes made have improved the article. I believe it's valuable work that should be published.

Response

Thank you. Much appreciated.

Reviewer 2 Report

Comments and Suggestions for Authors

In my opinion, the changes made have improved the article. I believe it's valuable work that should be published.

Author Response

(The authors gave the same response as above.)

Reviewer 3 Report

Comments and Suggestions for Authors

The authors have taken into account most of my remarks. However, my concerns about the implementation of stochastic frontier analysis (SFA) with a small sample remain.

While the authors acknowledge the limitation of a small sample - comprising only four municipalities observed over 16 years - the application of SFA in this context raises substantial methodological concerns. SFA typically requires sufficient cross-sectional variation to reliably distinguish inefficiency from statistical noise. With only four cross-sectional units, there is a significant risk that inefficiency effects are not separately identifiable, and estimates of efficiency scores may be highly unstable or driven by unit-specific idiosyncrasies rather than systematic differences.

Moreover, SFA relies on strong distributional assumptions about the inefficiency and noise terms, which are difficult to validate or test with such a limited number of municipalities. While the use of bootstrapping to mitigate efficiency losses in estimation is noted, it does not compensate for the fundamental issue of under-identification and limited generalizability. In fact, bootstrapped standard errors in this setting may give a false sense of robustness.

Given these concerns, the validity of the results—both in terms of internal consistency and external relevance—is questionable. I strongly recommend that the authors consider alternative methodological approaches more suited to small-sample analysis. These could include case study approaches or qualitative comparative analysis (QCA), which can provide richer contextual insight when sample sizes are small.

Alternatively, the authors should provide a stronger justification for the use of SFA under such constraints, including sensitivity tests and robustness checks to assess the stability of their findings across model specifications.

The authors should carefully check the accuracy of the text. There are several typographical and grammatical/stylistic errors. For example, on line 286: “…Thei findings…” — the word “Thei” should be corrected by removing the letter “i”. Also, on line 290: “…the auditor general…” — do the authors mean “generally” or “in general”?

Author Response

(The authors gave the same response as above.)

Round 3

Reviewer 1 Report

Comments and Suggestions for Authors

der authors, your manuscript has improved and is much clearer and structured.

you have answered my comments.

congrats on the efforts.

Reviewer 3 Report

Comments and Suggestions for Authors

The authors have sufficiently addressed my concerns.